

# Langmuir Turbulence in the Arctic Ocean: Insights From a Coupled Sea Ice –Wave Model

Aikaterini Tavri[1], Chris Horvat[1], Brodie Pearson[2], Guillaume Boutin[3], Anne Hansen[2], and Ara Lee[2]

[1]Brown University, Providence, RI, USA
[2]Oregon State University, Corvallis, OR, USA
[3]Nansen Environmental and Remote Sensing Center and Bjerknes Centre for Climate Research, Bergen, Norway

**Correspondence:** Aikaterini Tavri `aikaterini_tavri@brown.edu`

**Abstract.** Upper ocean mixing governs the vertical transport of heat, momentum, and tracers in the ocean surface boundary layer (OSBL), yet large-scale climate models often misrepresent its underlying processes, leading to significant uncertainty in sea ice and ocean predictions. Langmuir turbulence (LT) is one of the primary mechanisms of mixing in the open ocean and is generated by the interaction of wind stress and wave-induced Stokes drift. Observations have confirmed LT activity in leads, polynyas, and the marginal ice zone (MIZ), however its spatial and seasonal variability remains poorly constrained. In this study, we conduct the first Arctic-wide assessment of LT potential using a coupled sea ice–wave model that integrates neXtSIM and WAVEWATCH III. We analyze the spatiotemporal variability of LT by examining model-resolved turbulent dissipation and vertical kinetic energy within the OSBL. Our analysis reveals that LT potential is higher in the MIZ during melt and freeze-up, when partial sea ice cover allows intermittent wave propagation. Under these conditions, LT commonly coexists with wind-driven shear, forming a mixed-forcing regime that shapes upper-ocean energetics in response to evolving sea ice and wave states. Sea ice concentration and wind–wave alignment strongly influence the intensity and distribution of LT-driven mixing. On average, LT contributes roughly 15% of the total upper-ocean dissipation in the Arctic MIZ, with episodic wave-driven events during transitional ice periods doubling local mixing rates compared to wind-only conditions. This analysis highlights the energetic role of wave-induced mixing in the upper ocean, with potential implications for vertical momentum transport, mixed layer structure, and sea ice–ocean interactions in the Arctic.

## 1 Introduction

The Arctic Ocean has traditionally been considered a region of weak upper-ocean mixing, primarily due to extensive sea ice cover that insulates the ocean from atmospheric forcing and dissipates wave energy (Morison et al., 1985; Pinkel, 2005). Under these conditions, turbulent exchange in the ocean surface boundary layer (OSBL) remains strongly suppressed, and vertical mixing occurs only during sporadic shear-driven and convective events. In recent decades, the rapid decline in sea ice, marked by the loss of multiyear ice, earlier seasonal melt onset, and expansion of open water area, has increasingly exposed the Arctic Ocean to wind and wave forcing, fundamentally shifting the traditional view (Stopa et al., 2016; Armitage et al., 2017; Muilwijk et al., 2024). These changes have amplified air–sea momentum transfer (Rainville et al., 2011; Dosser and Rainville,



2016) and expanded the Marginal Ice Zone (MIZ), a transitional region characterized by discontinuous ice cover that enables
surface wave propagation and interaction with the floe field (Collins et al., 2018; Boutin et al., 2020).

Within the MIZ, surface gravity waves play a central role in mediating air–sea interaction. They modulate sea ice breakup and accelerate melt through enhanced mechanical stress and turbulent mixing (Thomson and Rogers, 2014; Thomson, 2022). Beyond direct wave breaking, surface waves also generate upper-ocean turbulence through Langmuir turbulence (LT), that develops when wind-forced shear aligns with wave-induced Stokes drift (Craik and Leibovich, 1976; Leibovich, 1983; McWilliams
et al., 1997). LT forms coherent, counter-rotating Langmuir cells that vertically redistribute heat, momentum, and tracers, (Skyllingstad and Denbo, 1995; D'Asaro, 2014; Kukulka et al., 2013; Gargett and Grosch, 2014), and it has emerged as a key regulator of mixed layer dynamics in the open ocean (Belcher et al., 2012; Yang et al., 2014). The absence of LT parameterizations in ocean general circulation models contributes to systematic biases in mixed layer depth and sea surface temperature, particularly in wind- and wave-active regions such as the Southern Ocean (Belcher et al., 2012; Li et al., 2019).

Large eddy simulation (LES) studies show that LT deepens the mixed layer and enhances vertical entrainment fluxes by up to an order of magnitude compared to shear-driven turbulence alone, while also moderately increasing momentum fluxes (McWilliams et al., 1997; Sullivan et al., 2007). Enhanced vertical mixing leads to elevated turbulent kinetic energy (TKE) and stronger entrainment across density interfaces (Polton and Belcher, 2007; Pearson et al., 2015; Ali et al., 2019). These insights have motivated the development of new LT parameterizations for large-scale models that incorporate wave–current interactions
and Stokes production terms (Van Roekel et al., 2012; Harcourt, 2015; Li and Fox-Kemper, 2017). Despite these advances, existing parameterizations have not been systematically evaluated in sea ice-covered regions, where the physical environment deviates substantially from typical open-ocean conditions due to the sea ice dynamics (McWilliams and Sullivan, 2000; Smyth et al., 2002; Brenner and Horvat, 2024).

In the Arctic, sea ice modifies upper ocean mixing dynamics in several ways. It limits wave fetch, alters the directional spread
of wave energy, and attenuates short-wavelength wave components - all of which reduce the magnitude and vertical extent of Stokes drift (Ardhuin et al., 2016, 2020; Li and Fox-Kemper, 2017). Meanwhile, ice motion and floe interactions introduce additional sources of surface shear and turbulence (Skyllingstad and Denbo, 2001). Observations show that waves propagate long distances under sea ice and significantly influence mixing near leads, polynyas, and the ice edge (Drucker et al., 2003; Kirillov et al., 2013; Horvat et al., 2020; Cooper et al., 2022). The presence of Langmuir cells in sea ice openings, confirms that
the LT mechanism remains active in ice-covered waters, albeit with intermittent occurrence and modified structure (Dethleff and Kempema, 2007; Voermans et al., 2019). Although prior studies on wave - ice interactions have primarily focused on mechanical breakup of sea ice (Collins et al., 2018; Squire, 2018), the turbulent mixing contributions of LT in sea ice covered regions remain largely unexplored. Recent modeling studies have incorporated wave - ice interactions to investigate localized upper-ocean mixing (Horvat et al., 2016; Manucharyan and Thompson, 2017; Cooper et al., 2022; Brenner and Horvat, 2024;
Lo Piccolo et al., 2024), but no study has yet conducted a basin-wide, systematic evaluation of LT mixing potential under realistic Arctic sea ice and wave conditions.

In this study, we use a coupled sea ice - wave model that combines the neXtSIM Lagrangian sea ice model (Rampal et al., 2016) with the WaveWatch III (WW3) spectral wave model (Tolman et al., 2009). This modeling framework resolves surface





Stokes drift, wave energy, and wind stress under evolving sea ice conditions. Using this information, we conduct an Arctic-wide assessment of LT potential under realistic wave - ice interactions. Our primary objective is to identify when and where LT mixing -supporting conditions emerge and persist, particularly in relation to seasonal sea ice advance and retreat. To capture the influence of wind - wave directional alignment, we compute both the standard (turbulent) Langmuir number $La_t$ and the projected Langmuir number $La_{\mathrm{proj}}$, which incorporates the cosine of the wind - Stokes drift angle to account for directional misalignment. Using these metrics, we map surface turbulent mixing regimes across space and time and estimate the seasonal evolution of TKE dissipation and vertical kinetic energy (VKE) associated with LT throughout the Arctic.

Section 2 describes the coupled model configuration and details the computation of LT-relevant parameters. Section 3 presents our results, beginning with the spatiotemporal variability of wind and wave forcing. We then map the prevalence of distinct turbulent mixing regimes and quantify seasonal and regional patterns in dissipation and VKE. Finally, we assess the influence of wind - wave misalignment on LT energetics, highlighting where projected Langmuir diagnostics diverge from canonical estimates. Section 4 discusses the implications of our findings for Arctic mixed layer dynamics and model development and outlines key limitations of the study.

## 2 Data and Methods

### 2.1 The neXtSIM-WaveWatch III Coupled Model

### 3 Model Configuration

We employ a fully coupled wave - sea ice modeling framework that integrates the spectral wave model WAVEWATCH III (WW3) with the Lagrangian neXt-generation Sea Ice Model (neXtSIM) using the OASIS-MCT coupler (Boutin et al., 2021). WW3 simulates wave attenuation due to sea ice using parameterizations for scattering, inelastic flexure, and under-ice friction. These processes respond dynamically to changes in sea ice concentration, thickness, and floe size distribution (Boutin et al., 2021). Sea ice morphology actively modulates Stokes drift profiles in WW3 by governing wave attenuation and reshaping the directional structure of the wave field (Squire, 2018). The neXtSIM model captures sea ice thermodynamics and dynamics. For this study, a key feature is its prognostic floe size distribution, which provides floe size information to WW3 and influences wave attenuation alongside sea ice concentration and thickness. Previous studies have evaluated this coupling in idealized and realistic settings (Boutin et al., 2018; Ardhuin et al., 2018), showing that the wave-affected area matches ICESat-2 observations (Boutin et al., 2022). Boutin et al. (2021) used this coupled system to investigate changes in sea ice dynamics and deformation, and we extend its application to the upper ocean mixing. Specifically, we quantify surface friction velocity and Stokes drift to derive LT metrics under varying sea ice and wave conditions. By resolving both the mechanical feedback from wave radiation stress (Longuet-Higgins and Stewart, 1962) and the impact of floe fragmentation on wave energy dissipation, the model provides a physically consistent framework for estimating LT mixing potential across the Arctic (Van Roekel et al., 2012; Li et al., 2017, 2019).





Our simulation spans the period 2018 - 2022 over a pan-Arctic domain with 25 km nominal resolution. The sea ice and wave models exchange information every 30 minutes, and output is recorded every 3 hours. NeXtSIM receives oceanic boundary conditions from the GLORYS12 reanalysis and uses ERA5 for atmospheric forcing (Hersbach et al., 2020). Both models use the same physical parameters as the REF configuration described in Boutin et al. (2022). Appendix A (Table A1) lists the wave, wind, sea ice, and surface ocean variables used in this study, either directly from model output or computed from established

physical relationships.

### 3.1 Surface Stress Partitioning and Wind–Wave Forcing

To characterize momentum input into the ocean mixed layer under partial ice cover, we compute an effective surface stress that accounts for both the stress transmitted through both sea ice and the direct wind stress over open water. Following the stress partitioning framework of Brenner et al. (2021), the net ocean surface stress is defined as an area-weighted combination of

ice–ocean and atmosphere–ocean stresses, scaled by the local sea ice concentration:

$$\tau_{ocn} = A\tau_{io} + (1 - A)\tau_{ao} \tag{1}$$

where $A$ is the sea ice concentration (0=open ocean, 1=fully ice-covered), and the direct atmosphere-ocean stress, is:

$$\tau_{ao} = \rho_a C_{ao} |\mathbf{u}_a| \mathbf{u}_a \tag{2}$$

with $\rho_a$ as air density, $\mathbf{u}_a$ the 10-m wind velocity, and $\mathbf{C}_{ao}$ the air–sea drag coefficient over open water. In our configuration,

the atmosphere–ocean stress is parameterized using Equation 2, while the ice–ocean stress ($\tau_{io}$) is diagnosed directly from the neXtSIM sea ice model, based on the momentum transfer between sea ice and the underlying ocean. Although the neXtSIM-WW3 model setup does not include interactive ocean dynamics, it captures the surface forcing of momentum pathways which drives turbulence in the OSBL.

Subsequently, we define the effective friction velocity $u_*$ which represents the shear strength at the ocean surface:

$$u_* = \sqrt{\frac{|\tau_{ocn}|}{\rho_o}}. \tag{3}$$

where $\rho_o$ is the density of seawater. It provides the fundamental scaling for wind-driven mixing processes.

In addition to wind shear, surface waves contribute momentum through Stokes drift, the net Lagrangian transport of water particles due to wave orbital motion. In WW3, the surface Stokes drift components (z = 0) are computed from the two-dimensional wave energy spectrum $F(k,\theta)$ as:

$$U_{sx}^{z=0} = g \int \int \frac{k^2 \cos(\theta)}{\sigma^2} F(k,\theta) \, d\theta \, dk, \tag{4}$$





and

$$U_{sy}^{z=0} = g \int \int \frac{k^2 \sin(\theta)}{\sigma^2} F(k,\theta) \, d\theta \, dk. \tag{5}$$

where $\sigma$ is the wave frequency, $k$ is the wave number and $\theta$ the propagation direction. These expressions define the eastward and northward components of the surface Stokes drift vector $u_s$. The effective friction velocity and surface Stokes drift

combined, provide the necessary inputs for evaluating and understanding Langmuir-related mixing under varying sea ice conditions.

### 3.2  Langmuir turbulence metrics in the Arctic

The Langmuir number ($La_t$) is a widely used parameter for quantifying the relative contributions of wind stress and wave-induced Stokes drift to upper ocean turbulence (McWilliams et al., 1997). It is defined as:

$$La_t = \sqrt{\frac{u_*}{u_{s(0)}}}, \tag{6}$$

where $u_*$ is the friction velocity derived from wind stress, and $u_{s(0)}$ is the surface Stokes drift magnitude. In the open ocean, typical values of $La_t$ range between 0.2 and 0.5 (Belcher et al., 2012), suggesting strong wave influence and active Langmuir circulation development, although $La_t$ can reach values near or above 1 when wave effects are weak and wind-driven processes dominate (McWilliams et al., 1997; Belcher et al., 2012). These ranges are consistent with results from LES

and field observations showing that stronger LT and deeper mixing are associated with lower $La_t$ (Harcourt, 2015). $La_t$ is thus widely used in ocean modeling as a diagnostic of upper-ocean mixing regimes and to inform turbulence parameterizations. Despite $La_t$ being a standard metric for LT mixing potential, recent studies have proposed refined formulations that better reflect realistic ocean forcing conditions, particularly when wind and wave directions are misaligned (Kukulka et al., 2010; Van Roekel et al., 2012; Li and Fox-Kemper, 2017). Directional mismatches between the Stokes drift and wind stress can

significantly reduce the effective wave contribution to Langmuir circulations, especially in the presence of swell, variable winds, or broad directional wave spectra. To account for this, we also compute the projected Langmuir number ($La_{\mathrm{proj}}$) as a supplementary diagnostic following (Van Roekel et al., 2012):

$$La_{\mathrm{proj}} = \sqrt{\frac{|\mathbf{u}_*|}{|\mathbf{u}_{s(0)}| \cos(\theta_{\mathrm{ww}})}}, \tag{7}$$

where $\theta_{\mathrm{ww}}$ is the angle between the wind stress and Stokes drift vectors. This formulation projects the Stokes drift onto the

wind direction, isolating the component that constructively interacts with wind-driven shear to generate Langmuir circulations. In theory, this projection should ideally account for the dynamic orientation of Langmuir cells ($\alpha_L$), which has been shown to vary between the wind and wave directions depending on wave age, stratification, and Coriolis effects (Van Roekel et al., 2012; Li et al., 2017). However, given the spatial resolution of our model ($\tilde{2}5$ km), we cannot resolve the vertical shear structure





required to estimate $(\alpha_L)$ reliably. We therefore adopt the simplifying assumption $\alpha_L = 0$, treating $\theta_{\mathrm{ww}}$ (the wind–wave angle)

as the effective misalignment. As a result, even $La_{\mathrm{proj}}$ may overestimate the suppression of LT in cases where the actual Langmuir cell alignment lies between wind and wave directions. Thus, $La_{\mathrm{proj}}$ functions as an upper-bound estimate for directional suppression of wave influence on LT, providing valuable insight into the sensitivity of LT potential to wind–wave misalignment under mixed forcing conditions.

### 3.2.1 Langmuir Turbulence Energetics in the Arctic

To evaluate the impact of LT on upper-ocean energetics, we compute three diagnostics: the enhancement factor $\mathcal{E}$, the turbulent dissipation rate $\varepsilon$, and the mixed-layer-averaged vertical velocity variance $\langle w'^2 \rangle_{H_{ML}}$. The LT enhancement factor $\mathcal{E}(La_x)$ quantifies the amplification of turbulence due to Langmuir circulation. This factor is derived as a function of the Langmuir number scalings $La_x$, following the formulation of (Li et al., 2019):

$$\mathcal{E}(La_x) \equiv \mathcal{E} = |\cos\theta_{ww}| \sqrt{1 + (c_1 La_x)^{-2} + (c_2 La_x)^{-4}} \tag{8}$$

where $\theta_{ww}$ is the angle between wind and Stokes drift directions and $c_1$ and $c_2$ are empirical constants. Following (Van Roekel et al., 2012), we adopt $c_1 = 3.1$, $c_2 = 5.7$, and $\theta_{ww} = 0$ for $La_t$, and $c_1 = 3.1$, $c_2 = 5.4$ for $La_{proj}$. The enhancement factor is used to scale the TKE dissipation rate, estimated as:

$$\varepsilon = \frac{u_*^3}{h} \mathcal{E}(La_x) \tag{9}$$

where $u_*$ is the friction velocity, $h$ is the MLD, and $\varepsilon$ represents the rate at which TKE is dissipated (W/kg). This formulation is

based on empirically derived scaling relationships between Langmuir number and turbulence dissipation, developed primarily through LES studies (Van Roekel et al., 2012; Li et al., 2016, 2019). These relationships form the basis for parameterizing wave-driven turbulence enhancement in ocean models, including implementations within the K-Profile Parameterization (KPP) scheme (Li and Fox-Kemper, 2017). This framework accounts for surface waves providing an additional energy source for turbulence in the upper ocean boundary layer, thereby increasing dissipation rates relative to wind-only forcing.

For additional calculations, we decompose the TKE dissipation into wind shear- and wave-driven contributions. Shear-driven dissipation is estimated as:

$$\varepsilon_{\mathrm{shear}} = u_*^3 / \mathrm{MLD} \tag{10}$$

representing turbulence generated purely by wind-induced surface stress. Wave-induced effects are incorporated using $\mathcal{E}(La_x)$. The total dissipation is therefore given by:

$$\varepsilon_{\mathrm{total}} = \varepsilon_{\mathrm{shear}} \cdot \mathcal{E}(La_x) \tag{11}$$

and the Langmuir-induced component is calculated as:

$$\varepsilon_{\mathrm{LT}} = \varepsilon_{\mathrm{total}} - \varepsilon_{\mathrm{shear}} = \varepsilon_{\mathrm{shear}}(\mathcal{E}(La_x) - 1) \tag{12}$$



This formulation ensures that wind- and wave-driven mixing are explicitly separated, with $\varepsilon_{\mathrm{LT}}$ capturing only the wave-enhanced turbulence component. Thus, $\varepsilon_{\mathrm{shear}}$ reflects baseline mixing from wind forcing alone, while $\varepsilon_{\mathrm{LT}}$ isolates the addi-
tional turbulent energy input attributable to wave–wind interactions via LT.

To estimate the vertical component of turbulent motions, we adopted the empirical relationship developed by (Van Roekel et al., 2012), which relates mixed-layer-averaged vertical velocity variance $\left\langle w'^2 \right\rangle_{H_{ML}}$ to the surface friction velocity, wind–wave alignment, and the Langmuir number:

$$\left\langle w'^2 \right\rangle_{H_{ML}} = 0.6 \left[ u_* \cos(\theta_{ww}) \right]^2 \left( 1 + (c_1 La_x)^{-2} + (c_2 La_x)^{-4} \right) \tag{13}$$

Here, $\left\langle w'^2 \right\rangle_{H_{ML}}$ is the vertical velocity variance averaged over the mixed layer, $u_*$ is the friction velocity, and $\theta_{ww}$ is the angle between wind and Stokes drift directions. The parameter $La_x$ is the Langmuir scaling, and $c_1$ and $c_2$ are the same empirical constants used in Equation 8, following Van Roekel et al. (2012). This definition accounts for the nonlinear amplification of vertical motions associated with LT, with enhancement increasing as $La_x$ decreases. To assess the influence of wave–ice interactions and wave–wind directional misalignment, we compute all energetic estimates using both $La_t$ and $La_{proj}$. To es-
timate dissipation and VKE from LT, we use MLD data available from the GLORYS12V1 reanalysis (produced by Mercator Ocean for Copernicus Marine Service), which provides daily, $0.08°$ resolution global ocean fields. The MLD is defined using a potential density threshold criterion ($\Delta\sigma_\theta = 0.03\,\mathrm{kg\,m}^{-3}$ from surface) (Copernicus Marine Service, 2020), consistent with standard practice and well suited for capturing seasonal and regional stratification associated with freshwater input in the Arctic Ocean. However, this is not a prognostic variable responding dynamically to surface wave or wind forcing within our coupled
model. Instead, it provides a diagnostic estimate of stratification depth, which we use to contextualize LT-driven turbulence under realistic hydrographic constraints.

**Table 1.** Summary of Langmuir Turbulence Diagnostics and Scalings

| Quantity | Definition | Purpose |
|---|---|---|
| Langmuir number ($La_t$) | $La_t = \sqrt{u_*/u_s}$ | Commonly used LT scaling |
| Projected Langmuir number ($La_{proj}$) | $La_{proj} = \sqrt{u_*/(u_s \cos\theta_{ww})}$ | Accounts for wave–wind misalignment |
| Enhancement factor ($\mathcal{E}_{La_t}$) | $\mathcal{E}_{La_t} = \cos\theta_{ww}\sqrt{1 + (c_1 La_t)^{-2} + (c_2 La_t)^{-4}}, \quad (\cos\theta_{ww} = 1)$ | Langmuir-induced mixing enhancement |
| Projected enhancement ($\mathcal{E}_{La_{proj}}$) | $\mathcal{E}_{La_{proj}} = \cos\theta_{ww}\sqrt{1 + (c_1 La_{proj})^{-2} + (c_2 La_{proj})^{-4}}$ | Enhanced mixing with misalignment |
| Turbulent Dissipation ($\varepsilon$) | $\varepsilon = \frac{u_*^3}{h} \times \mathcal{E}(La_x)$ | Shear- and Langmuir-driven dissipation |
| Normalized VKE ($La_t$) | $\frac{\langle w'^2 \rangle_{H_{ML}}}{[u_*]^2} = 0.6 \left[ 1 + (c_1 La_t)^{-2} + (c_2 La_t)^{-4} \right]$ | LT-driven enhancement of vertical motions |
| Normalized VKE ($La_{proj}$) | $\frac{\langle w'^2 \rangle_{H_{ML}}}{[u_* \cos(\theta_{ww})]^2} = 0.6 \left[ 1 + (c_1 La_{proj})^{-2} + (c_2 La_{proj})^{-4} \right]$ | LT-driven enhancement of vertical motions with misalignment |





## 4    Results

### 4.1    Spatiotemporal variability of surface forcing across the Arctic

We introduce a metric to assess how often local conditions under ice resemble those in open water (OW), allowing us to evaluate the occurrence and intensity of surface forcing relevant to LT in ice-covered regions. We focus on three key variables: surface friction velocity ($u_*$), surface Stokes drift velocity ($u_{s(0)}$), and significant wave height ($h_s$). For each grid cell $(x,y)$ and season, we compute the fraction of time steps during which a variable $X$ ($u_*$, $u_{s(0)}$, or $h_s$) in ice-covered conditions (SIC $\geq 0.15$) exceeds the seasonal median of the same variable calculated over OW conditions (SIC $< 0.15$) across the Arctic

domain. This results in a spatially resolved indicator of how often local forcing under sea ice reaches or surpasses levels typically observed in open water:

$$\text{OW\_Exceedance}_{\text{ice}}(x,y) = \frac{N_t\left[\text{SIC}_t(x,y) \geq 0.15 \wedge X_t(x,y) \geq X^{(s)}_{\text{median-OW}}\right]}{N_t\left[\text{SIC}_t(x,y) \geq 0.15\right]} \tag{14}$$

where $X^{(s)}_{\text{median-OW}}$ is the seasonal median computed over all open-water grid cells and $N_t[\cdot]$ counts the number of time steps satisfying the condition. Seasons were defined using meteorological groupings: DJF (Dec–Feb), MAM (Mar–May), JJA

(Jun–Aug), and SON (Sep–Nov), assigned based on the month of each time point in the dataset.

The distribution of $u_*$ is narrowly centered around $\sim 0.01 \; m/s$, with minimal seasonal variability (Figure 1a), suggesting relatively consistent wind-driven shear stress across the year in the OW regions. In contrast, $u_{s(0)}$ is centered around $\sim 0.1 \; m/s$ and exhibits stronger seasonality (Figure 1b), with elevated values during winter months and reduced values in summer. The $h_s$ distribution spans a broader dynamic range, with higher seasonal means and medians in winter (Figure 1c), reflecting

increased wave activity during this period. Figures 1d-f, show the spatial distribution of OW exceedance frequencies within the ice-covered Arctic (SIC $\geq 0.15$) . The exceedance map for $u_*$ reveals widespread regions, extending deep into the MIZ and central Arctic, where local shear often matches or exceeds OW medians. High exceedance rates are particularly pronounced in the Barents, Laptev, and Chukchi Seas and in Davis Strait. This indicates that atmospheric momentum can continue to penetrate efficiently through the sea ice cover across extensive areas of the Arctic. It supports the hypothesis proposed by

Martin et al. (2014, 2016), who argued that momentum transmission through sea ice can be regionally and seasonally enhanced when internal ice stress is low. In loosely packed - mobile ice, wind coupling to the ocean can remain strong or even be enhanced relative to OW. In contrast, exceedance frequencies for $u_{s(0)}$ are essentially zero, indicating that Stokes drift is heavily suppressed beneath sea ice due to wave attenuation and scattering processes (Ardhuin et al., 2016). This suppression is consistent with theoretical and observational studies showing that the Stokes transport rapidly decays under even partial

ice cover (Ardhuin et al., 2016; Herman, 2017; Liu and Mollo-Christensen, 1988). Despite the lack of $u_{s(0)}$ exceedance, $h_s$ shows notable exceedance frequencies, primarily concentrated along the seasonal ice edge. This suggests that surface waves are capable of penetrating into partially ice-covered areas, particularly near dynamic or retreating ice margins (Thomson et al., 2018). Localized wave presence, highlights the spatially constrained but temporally episodic nature of wave-driven processes





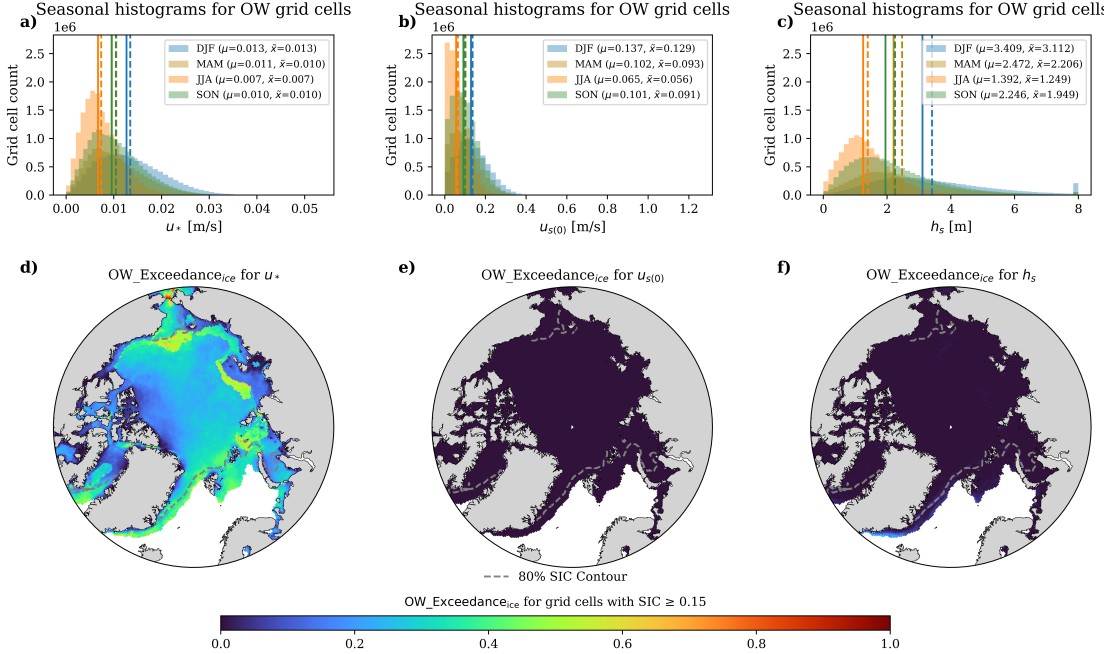

**Figure 1. Seasonal distribution and under-ice exceedance of surface friction velocity ($u_*$), surface Stokes velocity ($u_s$) and significant wave height ($h_s$) in the Arctic.** (a, b, c) Seasonal histograms of $u_*$, $u_{s(0)}$, and $h_s$ over OW grid cells ($SIC < 0.15$), with vertical lines marking the median (solid) and mean (dashed) for each season. (d, e, f) Fraction of time steps under ice-covered conditions ($SIC \geq 0.15$) where values of $u_*$, $u_{s(0)}$, and $h_s$ exceed the seasonal median open water threshold. The dashed contour represents the median 80% SIC boundary, indicating the typical MIZ extent. Color scale denotes the frequency of exceedance (0 to 1), with high values indicating regions of frequent open-water-like behavior in sea ice covered regions.

in the Arctic MIZ. The contrast between the widespread exceedance of $u_*$ and the confined nature of $h_s$ exceedance, illustrates

a key asymmetry in the surface forcing regime across the sea ice covered Arctic.

The spatial distribution and seasonal evolution of the median Langmuir number ($La_t$) characterize dominant upper-ocean turbulent mixing regimes across the Arctic (Figure 2). In all seasons, elevated median $La_t$ values ($> 1.5$) dominate the central ice-covered Arctic, indicating conditions where wave influence remains weak and surface shear primarily governs mixing. In contrast, lower median $La_t$ values ($< 0.45$), which signal dynamically significant LT, appear in narrow, seasonally evolving

bands along the MIZ. These LT-favorable zones track the climatological ice edge and expand in late summer and fall (Figure 2e), when open water exposure and increased wave activity enhance Stokes drift penetration. Spring and winter patterns (Figures 2b–c) exhibit sharp transitions between high and low $La_t$, showing how compact ice suppresses wave-driven mixing. The 5-year climatological median (Figure 2a) reveals a persistent band of low $La_t$ encircling the perennial ice pack and closely



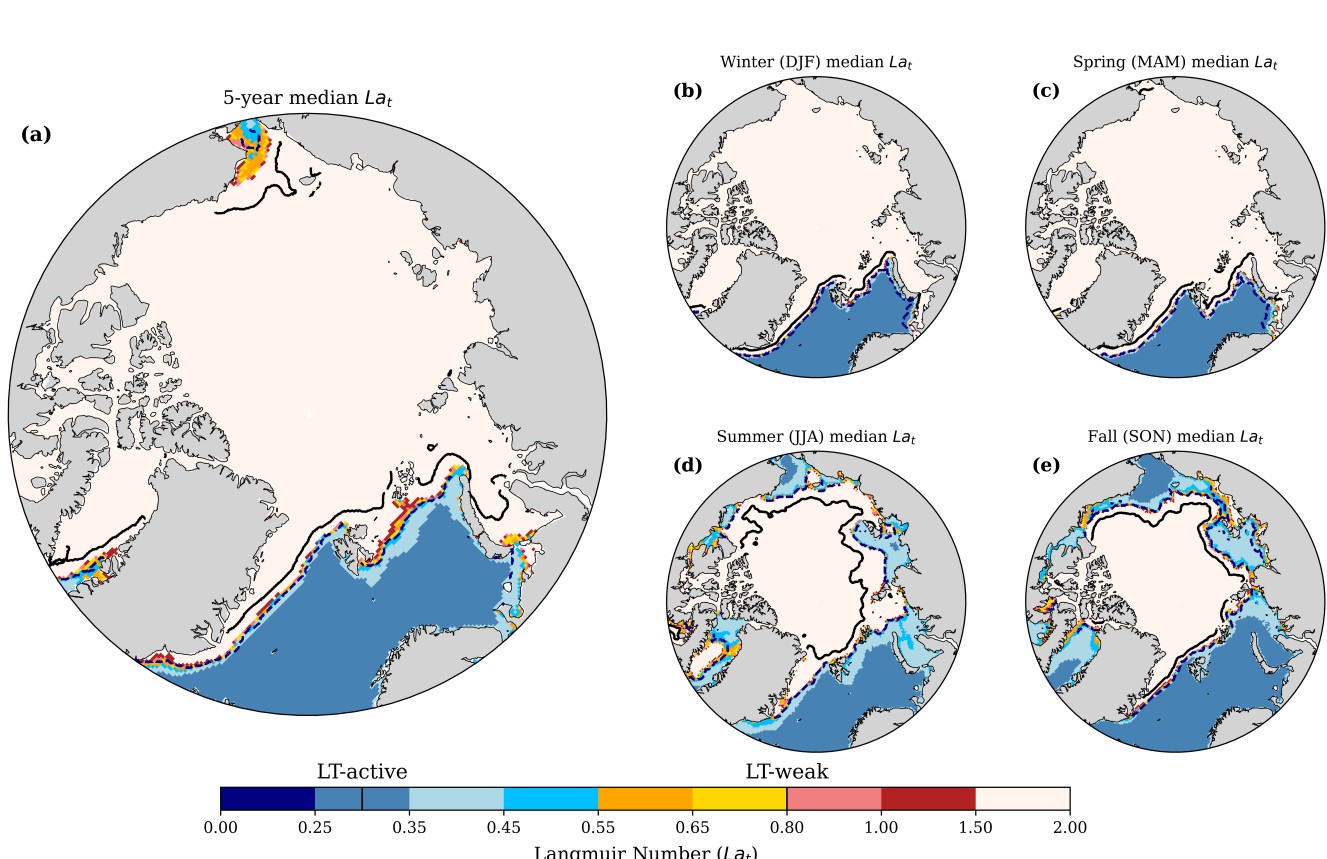

**Figure 2. Spatial distribution of the median Langmuir number $La_t$, integrated over 2018 to 2022, along with its seasonal medians.**
Panel (a) presents the overall median across all seasons. The median 15% and 80% SIC contours are overlaid in black and blue colors
marking the SIC-defined extent of the MIZ across seasons. Panels (b–e) show the medians for winter (DJF), spring (MAM), summer (JJA),
and fall (SON), respectively. The typical $La_t$-active open ocean and LT-weak thresholds are also indicated on the color scale for reference.

tracking the 15% SIC contour. This alignment, shows that there is LT potential in the seasonal MIZ, and indicates the critical
role of sea ice cover in modulating access to wave energy and, consequently, shaping the turbulent mixing regimes.

To further explain the seasonal controls on LT potential within the MIZ, we analyze the frequency of surface Stokes drift
($u_{s(0)}$) and friction velocity ($u_*$) exceedance across physically motivated thresholds. These thresholds correspond to varying
regimes of wave-induced mixing intensity, ranging from weak wave influence to conditions strongly conducive to LT. A day
is classified as a MIZ day if at least one grid cell satisfies the condition $0.15 \leq$ SIC $\leq 0.8$, ensuring that partial ice cover and
associated wave access are represented. Our findings, summarized in Figure A1, reveal that low-intensity surface Stokes drift
forcing ($u_{s(0)} > 0.02 \mathrm{~m\,s^{-1}}$) is ubiquitous in winter (99.8% of MIZ days) and spring (98%), and remains substantial even in
summer (71.7%) and fall (72.0%), showing that weak wave-driven turbulence is a persistent background feature of the MIZ,





even during seasons with reduced wave activity. Moderate wave forcing ($u_{s(0)} > 0.05 \text{ m s}^{-1}$), often associated with LT-active in prior studies, is similarly common in DJF (98%) and MAM (86%), but occurs on only 20.2% and 27.0% of MIZ days in

JJA and SON, respectively. The occurrence of strong wave forcing ($u_{s(0)} > 0.10 \text{ m s}^{-1}$), corresponding to $La_t \approx 0.3-0.4$ and indicative of enhanced LT potential, is seasonally skewed—observed on 85% of MIZ days in DJF and 58% in MAM, but only $\tilde{1}0\%$ in summer and fall. Extreme events ($u_{s(0)} > 0.20 \text{ m s}^{-1}$), characteristic of storm-driven swell penetration or near ice-edge exposures, are rare in JJA and SON (<10%), though reach up to 43% in DJF.

     Wind-driven shear emerges as a pervasive and seasonally consistent component of upper ocean forcing in the MIZ. Friction

velocities exceeding $0.01 \text{ m s}^{-1}$ occur on nearly all MIZ days throughout the year, indicating a persistent baseline for turbulence generation via wind stress. Moderate shear conditions ($u_* > 0.03 \text{ m s}^{-1}$) are sustained during 44% of winter days and 30% of spring days, but their prevalence diminishes sharply in summer and fall to approximately 10%. Strong wind forcing ($u_* > 0.04 \text{ m s}^{-1}$) remains infrequent, even in winter, where it occurs on just 6% of MIZ days. These patterns highlight a pronounced seasonal asymmetry in LT-relevant forcing, where wind stress offers a continuous, albeit variable, source of me-

chanical energy, while wave-driven enhancement is intermittent and strongly constrained by sea ice. The episodic nature of wave access, particularly during summer and autumn, restricts the occurrence of fully wave-dominated turbulence regimes. Even in OW, Langmuir numbers below 0.25, indicative of intense LT and wave-dominant mixing, are not common (Figure 2), aligning with phase-space constraints previously identified by Li et al. (2019). As a result, the prevailing state of turbulence within the MIZ is one of mixed forcing, characterized by intermediate Langmuir numbers ($0.35 \lesssim La_t \lesssim 1.5$). These values

suggest a regime where LT is present but not dynamically dominant.

### 4.2    Mapping upper ocean mixing regime dynamics in the Arctic

Several approaches have been developed to classify turbulence regimes in the OSBL, each emphasizing different regions of the non-dimensional parameter space defined by $La_t$. In the context of LT, LES studies have classified surface mixing regimes based on the relative contributions of wind shear and Stokes drift to TKE production and dissipation (e.g., Belcher et al.,

2012; Li et al., 2019). Following the framework of Li et al. (2019), we adopt regime boundaries to distinguish between wave-dominated, shear-dominated, and mixed-forcing conditions. At each grid cell location $(x, y)$ and analysis time $T$, the local Langmuir number $La_t(x, y, T)$ is used to assign a mixing regime:

$$\mathcal{R}(x,y,T) = \begin{cases} 1 & \text{if } La_t(x,y,T) > 0.94 \quad \text{(Shear-dominated)} \\ 2 & \text{if } 0.43 < La_t(x,y,T) \leq 0.94 \quad \text{(Mixed forcing)} \\ 3 & \text{if } La_t(x,y,T) \leq 0.43 \quad \text{(Wave-dominated)} \end{cases} \tag{15}$$

     To contextualize these regimes in relation to sea ice conditions, we define spatial regions $\Omega(T)$ at each time step $T$ based on

the sea ice concentration $\text{SIC}(x, y, T)$:





$$\Omega_{\text{ice}}(T) = \{(x,y)\,|\,\text{SIC}(x,y,T) > 0.8\} \tag{16}$$

$$\Omega_{\text{MIZ}}(T) = \{(x,y)\,|\,0.15 \le \text{SIC}(x,y,T) \le 0.8\} \tag{17}$$

$$\Omega_{\text{OW}}(T) = \{(x,y)\,|\,\text{SIC}(x,y,T) < 0.15\} \tag{18}$$

For each region $\Omega$ and regime $r \in \{1,2,3\}$, we compute the spatial fraction of grid cells classified in regime $r$ at time $T$ as:

$$f_r^{\Omega}(T) = \frac{1}{|\Omega(T)|} \sum_{(x,y)\in\Omega(T)} \delta\left(\mathcal{R}(x,y,T) = r\right) \tag{19}$$

where $\delta(\cdot)$ is the indicator function (equal to 1 when the condition is true, 0 otherwise), and $|\Omega(T)|$ is the number of valid grid cells in region $\Omega(T)$.

For the regime transitions, we track changes in $\mathcal{R}(x,y,T)$ from one time step to the next. The number of transitions from regime $r_n$ to $r_m$ at location $(x,y)$, restricted to grid cells within region $\Omega$, is given by:

$$T_{r_n \to r_m}^{\Omega}(x,y) = \sum_{T=2}^{T_{\max}} \delta\left(\mathcal{R}(x,y,T-1) = r_n\right) \cdot \delta\left(\mathcal{R}(x,y,T) = r_m\right) \cdot \delta\left((x,y) \in \Omega(T-1) \cap \Omega(T)\right) \tag{20}$$

To account for differences in temporal sampling coverage across the domain, we normalize the transition count by the number of time steps in which a grid cell lies within the region:

$$\bar{T}_{r_n \to r_m}^{\Omega}(x,y) = \frac{T_{r_n \to r_m}^{\Omega}(x,y)}{N_{\Omega}(x,y)} \tag{21}$$

where $N_{\Omega}(x,y)$ is the number of time steps in which $(x,y)$ satisfies the condition defining region $\Omega$. This normalization yields a dimensionless frequency in $[0,1]$ that reflects how often a specific regime transition occurs at each location over the analysis period.

Figure 3 shows how differences in wind and wave forcing, previously quantified using Equation 14, shape the dominant mixing regimes across the Arctic. In the open ocean, where waves are most accessible, $La_t$ values frequently fall below 0.43, indicating wave-dominated turbulence. In the MIZ, a transitional region with variable sea ice concentration, mixing regimes more often fall within the mixed-forcing range ($0.43 < La_t \le 0.94$). Consistent with the suppressed Stokes drift found under sea ice, the majority of the interior Arctic exhibits persistently high $La_t > 0.94$, indicating that shear-driven mixing is the prevailing regime. This is reflected in panel (a), where much of the central Arctic and parts of the seasonal ice pack are classified as shear-dominated over the five years of the analysis.

The seasonal cycle in panel (b), highlights that the balance between wind-driven and wave-driven turbulence is highly sensitive to sea ice concentration. Wave-influenced regimes grow in summer and early fall, as ice retreats and wave exposure increases, while shear-driven conditions dominate during winter and spring. Transitions between regimes, shown in panel (c),





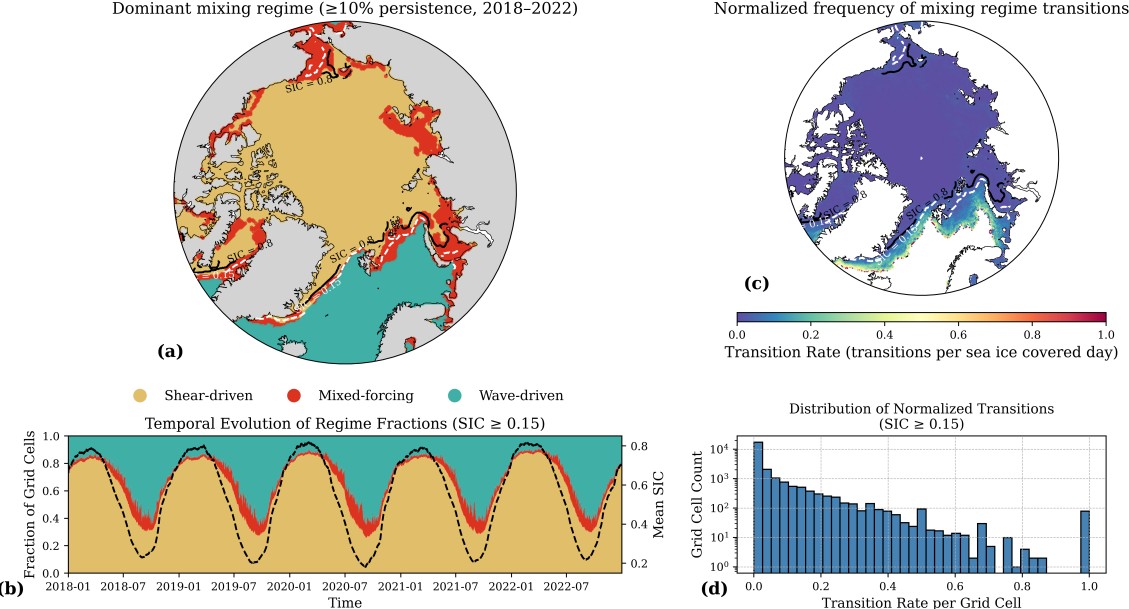

**Figure 3. Spatial and temporal characteristics of upper ocean mixing regimes in the Arctic.** Panel (a): Dominant mixing regime at each grid cell over 2018–2022, defined as the regime with at least 10% persistence during ice-covered periods (SIC ≥ 0.15). Panel (b): Temporal evolution of mixing regime fractions across all ice-covered grid cells, showing strong seasonal cycles in shear-driven (gold), mixed-forcing (red), and wave-driven (teal) conditions. Black dashed line indicates mean sea ice concentration. Panel (c): Spatial distribution of the normalized frequency of mixing regime transitions, computed as transitions per sea ice-covered day and panel (d): Histogram of normalized transition rates across grid cells with SIC ≥ 0.15.

occur most frequently near the ice edge and shelf boundaries-regions also identified earlier as hotspots of mixed wind- and wave- forcing. The rarity of transitions in the interior ice pack, reinforces the idea that sea ice acts as a stabilizing control on the upper ocean mixing regime. Panel (d) confirms that mixing regimes are persistent across perennial sea ice grid cells, however, there is a non-negligible subset, where turbulent forcing is highly variable.

Figure 4 provides further insight into how local and spatially aggregated values of the $La_t$ vary as a function of SIC, helping to contextualize the spatial patterns of each mixing regime. The left panel shows that median $La_t$ values increase monotonically with SIC across all spatial scales, indicating a systematic suppression of wave-driven mixing under increasing ice cover. While the lowest $La_t$ values ($< 0.43$), associated with strong LT, are found only in open water or sparsely ice-covered conditions, values exceeding 0.94 dominate in areas with moderate to high SIC, consistent with earlier findings that wave forcing is effectively attenuated by even partial ice coverage. Panel (b) illustrates the pronounced subgrid variability of $La_t$ using histograms of the minimum, mean, and maximum values computed over local $3 \times 3$ grid cell neighborhoods.





We defined these neighborhood metrics to provide physical insight into the surrounding dynamical environment experienced by each MIZ grid cell. Specifically, $La_t^{\min}$ identifies the most wave-affected neighboring point, serving as an upper bound on local wave influence, while $La_t^{\max}$ reflects the most shear-dominated condition in the vicinity. The spatial mean ($La_t^{\mean}$) yields a smoother representation that closely tracks the local (single grid cell) $La_t$, highlighting dominant regime tendencies. In contrast, the extremes reveal localized deviations that are masked by coarser averages. From this interpretation we realize that mixed-forcing regimes in the MIZ are not only seasonally modulated but also sensitive to spatial scale, suggesting a potential need for subgrid-aware approaches in OSBL parameterizations under partial ice cover.

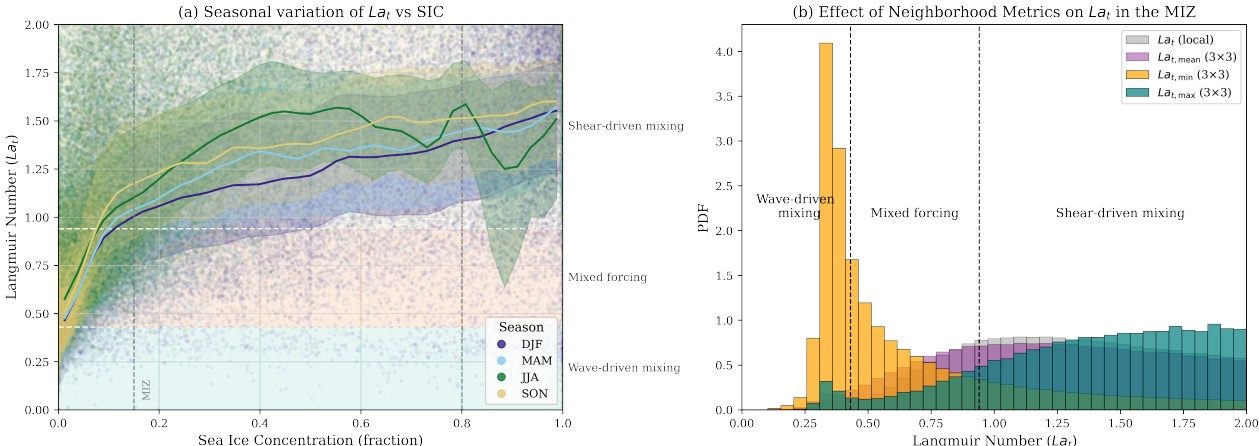

**Figure 4. Sea ice concentration dependence of the turbulent Langmuir number.** (**Left**) Parametric plot of $La_t$ against binned sea ice concentration (SIC). Solid lines denote the median $La_t$ for each season and bin, with shading indicating inter-quartile ranges. The white dashed lines mark the wave-dominated ($La_t < 0.43$) and shear-dominated ($La_t > 0.94$) regime thresholds. (**Right**) Normalized histogram of $La_t$ values in each smoothing category, aggregated over all ice-covered grid cells (SIC $\geq 0.15$). This histogram highlights the asymmetry and variability in the spatial scale and intensity of Langmuir turbulence across the Arctic.

### 4.3 Pan-Arctic dissipation rates and normalized VKE

This section highlights the energetic role of LT in the Arctic beyond spatial regime classification. We examine the dissipation rate $\varepsilon$ and its enhancement relative to the expected dissipation if only shear-driven mixing were present. Figure 5a displays the pan-Arctic mean dissipation rate over the 2018–2022 period. Elevated $\varepsilon$ values are found along the climatological MIZ and in marginal seas, where wave–ice interactions persist seasonally. Dissipation hotspots exceed $\varepsilon \approx 3 \times 10^{-7} \text{ m}^2 \text{s}^{-3}$, aligning with in-situ observations of enhanced turbulent mixing in wave-active polynyas and shelf seas (Rippeth and Fine, 2022).

Additionally, we compute the normalized dissipation ratio $\varepsilon_{\mathrm{LT}}/\varepsilon_{\mathrm{shear}}$ (figure 5b), which exhibits a strong inverse dependence on the Langmuir number $La_t$, consistent with theoretical and LES-based scaling relationships (Grant and Belcher, 2009; Li et al., 2019). Specifically, the theoretical scaling $\varepsilon_{\mathrm{LT}}/\varepsilon_{\mathrm{shear}} \propto 1/La_t^2$, derived from idealized mixed-layer turbulence theory, is included for reference and broadly captures the observed trend. Enhancement ratios exceeding 2 are observed at low $La_t$





($< 0.5$), indicating significant Langmuir-induced mixing. However, the observed enhancement remains below the theoretical limit, reflecting the modulating effects of stratification and sea ice cover. These high ratios primarily occur in low-SIC and OW regions, while compact sea ice strongly suppresses LT contributions. Seasonal histograms of spatially averaged dissipation (Figure 5c) reveal significantly higher values during September compared to March, reflecting both enhanced wave activity and reduced sea ice extent, favoring LT generation. This seasonal amplification is consistent with LES results that show LT

strengthening during low-stratification and high-wave periods (**?**).

Panels (d) and (e) further quantify the role of Lusing two complementary diagnostics. Figure 5d shows the spatially averaged Langmuir enhancement factor, $\mathcal{E}(\mathcal{L}\dashv_{\S}) - 1$, partitioned by sea ice concentration regime. This metric reflects the potential amplification of turbulence due to wave–wind interactions when Langmuir processes are active. The enhancement is strongest in open water ($\sim 0.45$), moderate in the marginal ice zone (MIZ; $\sim 0.15$), and negligible under compact sea ice. In contrast,

Figure 5e presents the time-weighted fractional contribution of Langmuir-induced dissipation to the total dissipation budget. This panel captures the realized impact of LT over the total simulation period by integrating both wind- and wave-driven components across time. Despite low $La_t$ values indicating wave-dominant conditions, the actual dissipation may remain shear-dominated due to intermittent high-wind events or periods when LT is suppressed, such as under partial or full ice cover.

To complement the pan-Arctic assessment, we analyze the temporal evolution of Langmuir-driven mixing at two dynami-

cally contrasting sites using 100 km-radius kernel averages centered in the Central Arctic (84.5°N, 202.0°E) and the Barents Sea (78.0°N, 43.9°E). Figure 6 presents the 2022 time series of turbulent dissipation rate ($\varepsilon$) and mixed-layer-averaged vertical kinetic energy ($\langle w'^2 \rangle_{H_{\mathrm{ML}}}$), alongside daily sea ice concentration (SIC). At both locations, LT-enhanced metrics exhibit pronounced seasonal variability, tightly coupled to wave exposure modulated by evolving ice conditions.

In the Central Arctic, LT-enhanced dissipation rates ($\varepsilon_{\mathrm{LT}}$) closely track the shear-only estimates but consistently exceed them,

including during periods of compact ice cover. Seasonal peaks emerge in January and April-coinciding with intensified wind forcing- and again in October following a mid-summer plateau. This persistent exceedance suggests sustained LT potential even under high SIC, likely due to mobile or loosely packed conditions that permit momentum transfer. Additionally, we observe a peak in dissipation in July, as SIC starts decreasing. In contrast, VKE shows less regular seasonality but exhibits distinct peaks in end of March/April and December that surpass the dissipation rate, implying episodic energy injections. These events likely

reflect storm-driven activity under high SIC, where wind and wave forcing remain effective despite partial ice cover.

The Barents Sea displays a different evolution, shaped by substantial springtime ice retreat and transition to OW by early summer. This shift is accompanied by sharp increases in VKE during April-May, consistent with enhanced LT potential during early retreat sea ice phases. Throughout the year, LT-enhanced dissipation remains elevated above the shear-only baseline, with particularly strong signals in fall and winter. Compared to the Central Arctic, active LT conditions occur earlier and persists

longer, facilitated by a deeper climatological mixed layer and generally lower SIC. Interestingly, VKE diminishes as SIC drops toward open-water conditions-a pattern suggesting that LT activity in this marginal setting peaks under intermediate SIC ($\sim 0.4$–$0.7$), where wave penetration, ice damping, and wind–wave alignment optimally enhance vertical energy. This seasonal decline likely reflects increasing upper-ocean stratification due to surface heating and sea ice melt, which suppresses vertical motions despite low sea ice concentration. The disappearance of VKE highlights the sensitivity of Langmuir-driven turbulence



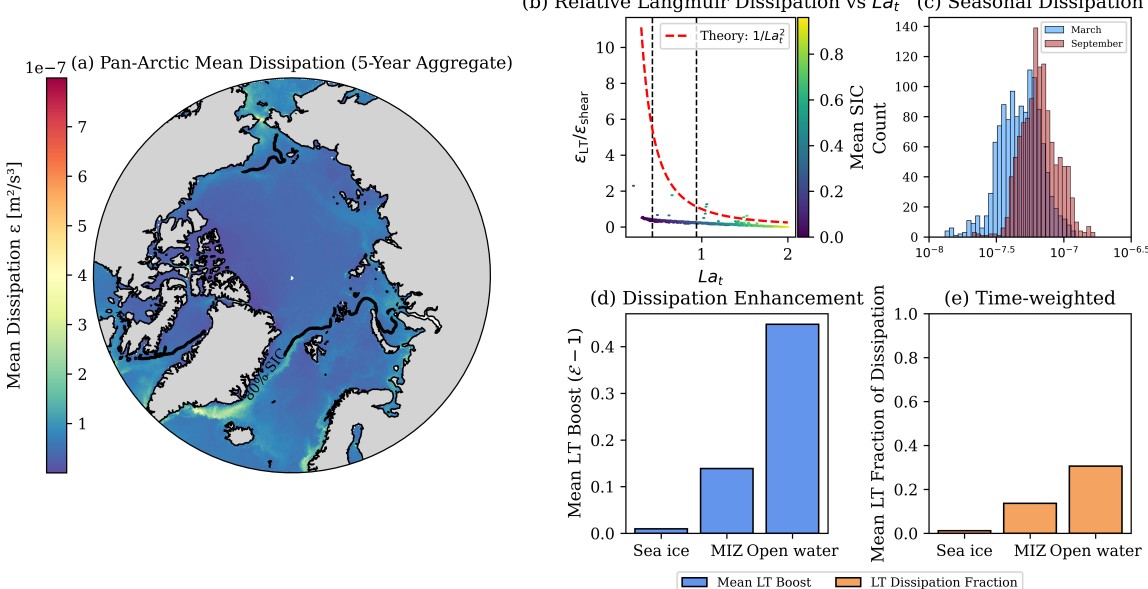

**Figure 5.** Spatial and regime-dependent structure of upper ocean turbulent dissipation across the Arctic. (a) Mean dissipation rate ($\varepsilon$) over the 2018–2022 period, with 80% sea ice concentration (SIC) contour shown in black. (b) Relative contribution of LT to total OSBL dissipation ($\varepsilon_{\mathrm{LT}}/\varepsilon_{\mathrm{shear}}$) as a function of $La_t$, colored by mean SIC. Dashed lines denote $La_t$ regime thresholds for wave-driven, mixed and shear driven mixing. The red dashed line shows the theoretical scaling $\varepsilon_{\mathrm{LT}}/\varepsilon_{\mathrm{shear}} \propto 1/La_t^2$, derived from idealized mixed-layer turbulence theory (Grant and Belcher, 2009). (c) Seasonal histogram of domain-averaged dissipation for March and September, indicating an annual modulation linked to sea ice retreat and increased wave exposure. (d) Mean dissipation enhancement due to Langmuir turbulence, quantified as $\varepsilon/\varepsilon_{\mathrm{shear}} - 1$, across sea ice regimes, illustrating the increasing energetic importance of LT in low-SIC conditions. (e) Mean fractional contribution of Langmuir-driven dissipation across SIC regimes, time-weighted over all valid time steps, emphasizing the growing role of LT in OW regions relative to compact ice.

to evolving wave conditions and buoyancy structure, consistent with theoretical expectations and LES results (Van Roekel et al., 2012; Li et al., 2019).

    Together, the 2022 time series demonstrate that LT potential in both regions is strongly modulated by SIC and wave forcing, but not limited to OW conditions. Even under compact ice, enhanced dissipation and elevated VKE imply that wind-wave momentum coupling remains dynamically active. The co-variation of $\varepsilon_{\mathrm{LT}}$ and $\langle w'^2 \rangle_{H_{\mathrm{ML}}}$ confirms that Langmuir processes

contribute not only to enhanced dissipation but also to sustained vertical momentum transfer throughout the year. These findings reinforce the spatial patterns identified in Figure 5, highlighting the importance of representing LT variability when modeling upper-ocean mixing in seasonally ice-covered environments.





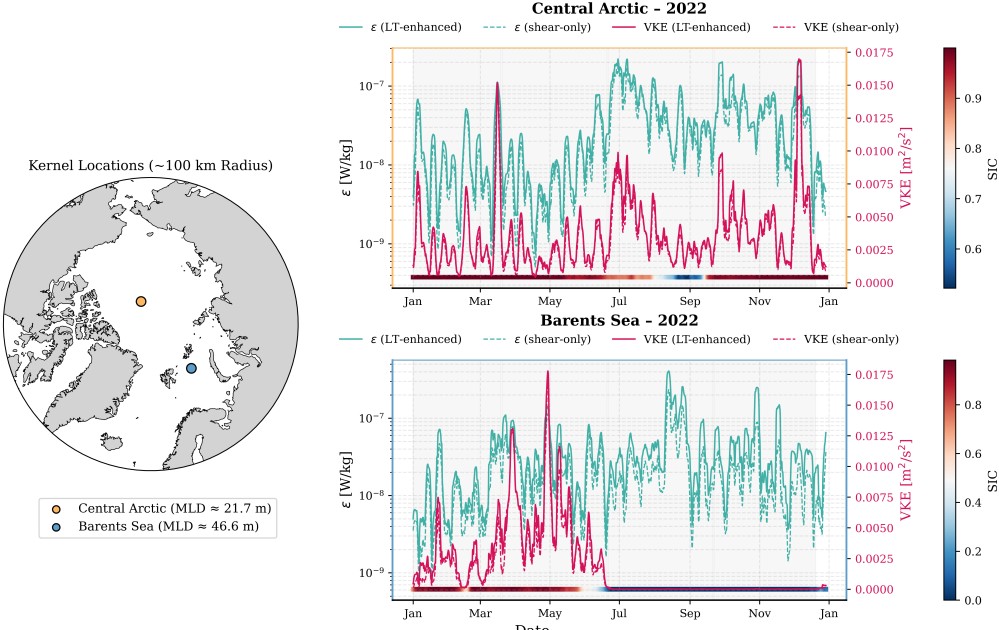

**Figure 6. Langmuir turbulence diagnostics for kernel regions** ( Central Arctic and Barents Sea), each defined as a 5×5 gridcell (∼100 km radius) area. **Left**: Map of kernel locations. **Right**: For each region, time series of Langmuir turbulence dissipation ($\epsilon_{LT}$), normalised vertical kinetic energy (VKE), and SIC for a year (2022).

## 4.4 Impact of Wind–Wave Misalignment on Langmuir Turbulence-Induced Dissipation in the Arctic

In this section, we compute the dissipation ratio $\varepsilon_{\mathrm{La_{proj}}}/\varepsilon_{\mathrm{La_t}}$ to assess how directional misalignment between wind and waves
modulates the efficiency of Langmuir-driven mixing. This ratio compares two formulations of the Langmuir number: one incorporating projection onto the wind direction ($La_{\mathrm{proj}}$) and one assuming perfect alignment ($La_t$). Both formulations use the same enhancement framework described in Section 2.2.3. Figure 7 visualizes the dissipation ratio $\varepsilon_{La_{\mathrm{proj}}}/\varepsilon_{La_t}$ and quantifies the modulation from misalignment, revealing moderate suppression across the interior ice pack, typically in the 0.7–0.9 range. This suggests that LT remains active but is somewhat diminished due to wind–wave misalignment under high sea ice
concentration.

The VKE difference map further supports this interpretation, showing $\Delta\mathrm{VKE} \approx 0$ across much of the central Arctic. This implies that Langmuir-induced vertical motions are minimal in these regions, regardless of whether wind–wave misalignment is explicitly accounted for. Rather than directional decorrelation, it is the persistent presence of compact sea ice that primarily suppresses the Stokes drift and surface wave input needed to sustain LT. Consequently, both the standard enhancement factor $\mathcal{E}_{La_t}$ and its misalignment-corrected counterpart $\mathcal{E}_{La_{\mathrm{proj}}}$ yield similarly low estimates of vertical kinetic energy in these



regimes. This highlights that dense sea ice—not misalignment—is the dominant limiting factor for LT activity in the central Arctic.

In particular, the dissipation formulation includes an explicit $|\cos(\theta_{\mathrm{ww}})|$ scaling term in addition to the projected Langmuir number, whereas for the VKE expression, the angle dependence is absorbed into the scaling via $La_{\mathrm{proj}}$. As a result, VKE differ-
ences are more spatially uniform and weaker than the mapped dissipation ratio. This structural distinction between dissipation and VKE parameterizations likely contributes to the stronger regional variability observed in dissipation ratios compared to VKE. In panel (c), the binned relationship between dissipation ratio and $\cos(\theta_{\mathrm{ww}})$ reveals a near-linear dependence. As alignment improves ($\cos(\theta_{\mathrm{ww}}) \to 1$), the dissipation ratio approaches unity, while strong misalignment ($\cos(\theta_{\mathrm{ww}}) < 0.6$) results in up to 50% suppression of wave-enhanced dissipation. Colored by SIC, the scatterplot confirms that misalignment effects are
most pronounced in partially ice-covered waters, where directional variability is high and LT remains active.

Interestingly, the MIZ exhibits only modest differences between $La_t$- and $La_{\mathrm{proj}}$-based dissipation estimates, with mean dissipation ratios around 0.8. This relatively weak sensitivity to misalignment in the MIZ may arise from several factors. First, both wind and wave fields in the MIZ are typically more energetic and spatially coherent than in the interior ice pack, reducing the likelihood of extreme directional decoupling (Stopa et al., 2018). Second, local wave–wind climatologies in the
MIZ may already reflect partial alignment due to fetch-limited wave growth under intermediate ice concentrations. Third, under strong turbulence forcing, the nonlinear response of Langmuir enhancement factors to $La_t$ or $La_{\mathrm{proj}}$ may saturate, such that moderate misalignment produces only limited suppression. Hence, these factors suggest that in the MIZ, projected Stokes–wind alignment (as captured in $La_{\mathrm{proj}}$) introduces only a secondary correction to $La_t$-based dissipation estimates, supporting the use of simpler diagnostics for bulk assessments while highlighting the added value of alignment-aware metrics in regional and
dynamic ice-covered regimes.

## 5   Discussion and Conclusions

Our results provide a comprehensive assessment of the spatiotemporal variability of Langmuir turbulence (LT) in the Arctic Ocean, revealing that LT is dynamically modulated by the interplay between wind stress, wave accessibility, and sea ice cover. Rather than existing as distinct binary states of wave- or shear-dominance, Arctic upper-ocean turbulence frequently manifests
as a mixed-forcing regime, particularly in the MIZ, where both mechanisms co-occur with varying intensities. Elevated surface shear stress across the MIZ and into the interior pack indicates that wind forcing alone is frequently sufficient to support LT, provided wave conditions permit. However, Stokes drift penetration, governed by sea ice concentration (SIC), modulates the degree to which wave-driven turbulence can be realized. This asymmetry yields a regime where wind-driven shear is often present, but wave forcing is spatially and temporally constrained, creating conditions favorable for intermittent but energetically
significant LT events.

The seasonal expansion of LT-favorable zones during late summer and early fall coincides with enhanced wave exposure and reduced SIC, producing stronger LT signals during these periods. Spring and winter medians, by contrast, reveal sharply bounded $La_t$ transitions that reflect suppression under compact ice. These are most pronounced in the MIZ, where SIC and





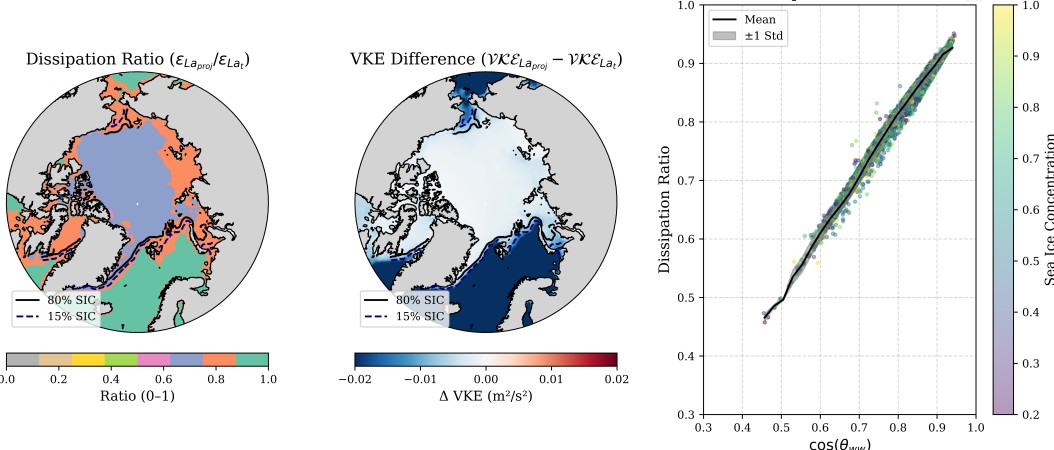

**Figure 7. Sensitivity of Langmuir-driven dissipation and VKE to wave–wind misalignment.** (a) Spatial distribution of dissipation ratio $\varepsilon_{\mathrm{La}_{\mathrm{proj}}}/\varepsilon_{\mathrm{La}_t}$, quantifying the suppression of dissipation due to misalignment. (b) VKE difference ($\Delta\mathrm{VKE} = \mathrm{VKE}_{\mathrm{La}_{\mathrm{proj}}} - \mathrm{VKE}_{\mathrm{La}_t}$), highlighting negative anomalies across the ice pack. (c) Dissipation ratio as a function of directional alignment ($\cos(\theta_{\mathrm{WW}})$), with point color indicating local SIC. Shaded band shows $\pm 1\sigma$ about the mean in each bin.

wave energy vary over short distances, leading to rapid shifts in mixing regime. Our classification framework confirms that

pure wave-dominant conditions are rare; instead, mixed-forcing regimes dominate across the MIZ. These emerge under partial ice cover, moderate SIC, and episodic wave events, suggesting that the MIZ functions as a responsive and sensitive transition zone between open-water and ice-covered states.

Scale-based diagnostics diagnostics reinforce this interpretation. Histogram-based kernel analyses of $La_t$ reveal that while gridcell means suggest shear dominance, neighboring extremes often reach wave-favorable values ($La_t^{\min} < 0.43$), indicating

that turbulence regimes are highly heterogeneous at the subgrid scale. Such variability, driven by SIC and directional wave fields can induce intense but spatially confined LT activity, particularly in the MIZ. These results highlight a key limitation of coarse-resolution (25 km) models, where grid-mean diagnostics may obscure critical small-scale variability in surface forcing that modulates mixing regime dynamics.

Dissipation and vertical kinetic energy (VKE) estimates further substantiate these findings. Langmuir-enhanced dissipation is

elevated in the MIZ and open-water sectors, with peak values exceeding $\varepsilon \sim 3\times 10^{-7}\,\mathrm{m}^2\,\mathrm{s}^{-3}$, consistent with prior observations in wave-active polynyas and marginal seas. In the Central Arctic, however, dissipation is generally lower and more seasonally muted, although transient increases in VKE and $\epsilon$ coincide with brief periods of moderate SIC. These results suggest that LT in compact ice is limited but not negligible, potentially driven by residual swell energy or directional wave refraction.

We also examined the influence of wind–wave misalignment on LT diagnostics using the projected Langmuir number $La_{\mathrm{proj}}$.

Arctic-wide maps show that incorporating misalignment reduces LT-driven dissipation and VKE in the central basin, with





suppressions exceeding 40% in some areas. However, in the MIZ, the impact of misalignment was notably modest, with dissipation ratios ($\varepsilon_{La_{\mathrm{proj}}}/\varepsilon_{La_t}$) close to 0.8–0.9, and minimal VKE reductions. This suggests that in dynamic, partially ice-covered environments where waves and winds are both active, the alignment angle may already be implicitly constrained by regional climatologies, or that the absolute magnitude of forcing dominates the response. This result supports the use of $La_t$ for

diagnosing broad regional patterns, while highlighting the added value of alignment-aware metrics in regional, more dynamic regions.

Concluding, our results show that LT in the Arctic varies in space and time, responding dynamically to changes in waves, sea ice, and wind. The MIZ emerges as the most sensitive zone for transitions between mixing regimes. While standard Langmuir number diagnostics capture broad patterns, they tend to overestimate turbulence under compact ice and underestimate it in

the MIZ, where mixed forcing dominates. Incorporating wind–wave misalignment improves realism but adds only a modest correction in the MIZ. As Arctic sea ice continues to thin and retreat, LT and mixed-forcing conditions are likely to expand, underlining the need for regime-aware turbulence parameterizations to simulate upper-ocean structure and air–ice–ocean feedbacks under climate change.

### 5.1 Limitations and Future Directions

Our approach is based on bulk diagnostics and empirical scalings, and thus carries several limitations. The use of 25 km resolution fields limits our ability to resolve fine-scale processes such as eddies, floe-scale wave attenuation, and narrow leads and polynyas, all of which modulate LT and upper-ocean mixing. Although local kernel statistics help capture some subgrid variability, direct evaluation of small-scale LT dynamics requires high-resolution modeling and in situ observations. Additionally, our analysis uses a coupled sea ice-wave simulation where mixed-layer depth is not dynamically responsive to wind–wave

forcing, without explicit ocean stratification or feedbacks. This constrains the accuracy of vertical mixing diagnostics and highlights the need for fully coupled ocean–ice–wave models to more realistically simulate LT evolution in the Arctic.

Future work should focus on integrating directional wave spectra and wind–wave coupling from fully coupled ocean–wave–ice models, allowing for a more explicit treatment of misalignment effects and wave propagation under ice. Observational efforts, including coordinated field campaigns using autonomous profilers, SWIFT drifters, and satellite altimetry, are also critical for

validating LT parameterizations in ice-covered waters. In particular, future studies should prioritize the MIZ, where our results suggest that mixed forcing dominates and where wave–ice–wind interactions are most dynamically active. Finally, evaluating the impact of LT on vertical tracer transport, stratification erosion, and ice–ocean heat exchange within climate models will be essential to fully quantify the role of LT in the evolving Arctic system.





**Appendix A**

**Table A1.** Definitions and descriptions of available parameters from neXtSIM and WW3 models.

| Parameter | Description | Units |
|---|---|---|
| **neXtSIM Model Parameters** | | |
| $\tau_x$ | Eastward stress at ocean surface | Pa |
| $\tau_y$ | Northward stress at ocean surface | Pa |
| $u_{\text{ice}}$ | Sea ice x velocity | $\mathrm{m\,s^{-1}}$ |
| $v_{\text{ice}}$ | Sea ice y velocity | $\mathrm{m\,s^{-1}}$ |
| $SIC$ | Sea ice concentration | - |
| $SIC_{\text{young}}$ | Young ice concentration | - |
| $SIT_{\text{young}}$ | Young ice thickness | m |
| $SIT$ | Sea ice thickness | m |
| $SNT$ | Surface snow thickness | m |
| $D_{\text{max}}$ | Maximum floe size | m |
| $D_{\text{mean}}$ | Mean floe size | m |
| $SSS$ | Sea surface salinity | PSU |
| $SST$ | Sea surface temperature | °C |
| $T_{2\text{m}}$ | 2-m air temperature | °C |
| **WW3 Model Parameters** | | |
| $dir$ | Sea surface wave from direction | degrees |
| $d_p$ | Sea surface wave peak direction | degrees |
| $f_p$ | Sea surface wave peak frequency | Hz |
| $H_s$ | Significant wave height | m |
| $ic1$ | Ice thickness | m |
| $ic3$ | Average ice floe diameter | m |
| $ice$ | Sea ice area fraction | - |
| $t_{02}$ | Mean wave period (from second moment) | s |
| $tus$ | Northward Stokes transport | - |
| $us$ | Northward surface Stokes drift | - |
| $wnd$ | wind speed | $\mathrm{m\,s^{-1}}$ |




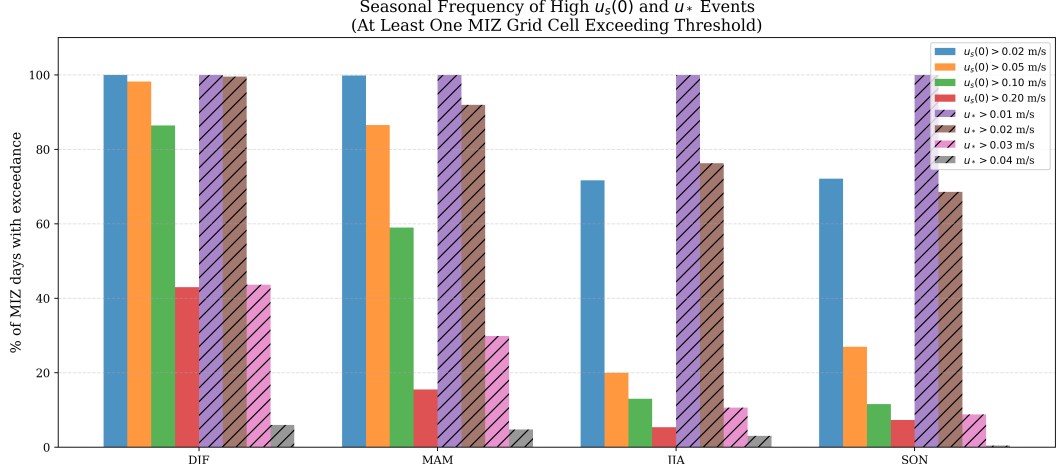

**Figure A1. Seasonal bar plots of high Stokes drift velocities ($u_{s(0)}$) and surface friction velocities ($u_*$) in the MIZ.**

*Data availability.* The model outputs and post-processed Langmuir turbulence diagnostics used in this study are available from the corresponding author upon request. ERA5 atmospheric reanalysis data are publicly available from the Copernicus Climate Data Store and the GLORYS12V1 ocean reanalysis is available through the Copernicus Marine Service.

*Author contributions.* writing—original draft preparation AT; conceptualization review and editing CH, BP and AT; model data curation and edits GB; early analysis and editing AH and AK;


*Competing interests.* The authors declare no conflicts of interest.

*Acknowledgements.* This research was supported in part by the National Science Foundation (NSF OPP-2146910 and OCE-2148655) and by Schmidt Sciences, LLC through the SASIP project.The simulations were performed on resources provided by Sigma2 - the National Infrastructure for High-Performance Computing and Data Storage in Norway.



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
