# Peer review of "Langmuir Turbulence in the Arctic Ocean: Insights From a Coupled Sea Ice –Wave Model"

_EGUsphere, 2025_

## Author Comment (AC1)

**Response to Reviewer #1**

Manuscript: Langmuir Turbulence in the Arctic Ocean: Insights From a Coupled Sea Ice-Wave Model

Authors: Aikaterini Tavri et al.

**General Comments**

**Author Response:**

We thank the reviewer for their constructive and detailed feedback. Below we provide responses to all general and specific comments. We have carefully revised the manuscript to address the issues raised, improved the explanation of turbulent kinetic energy scaling, and clarified the role of sea-ice.

**General Comment 1**

**Reviewer Comment:**

My first concern is on the use of an enhancement factor defined in Eq. (8) to scale the enhancement of TKE dissipation by Langmuir turbulence in Eq. (9). The Langmuir enhancement factor in Eq. (8) describes the enhancement of turbulent velocity scale, which is based on scalings of vertical velocity variance in a set of large eddy simulations described in Van Roekel et al., 2012. The TKE dissipation does not necessarily scale in the same way. In fact, it shouldn't scale the same way as it depends on the turbulent velocity scale cubed. One may instead use Eq. (5) in Belcher et al., 2012 to estimate the enhancement of TKE dissipation due to Langmuir turbulence. But I'm not sure it is possible to clearly attribute the TKE dissipation to shear-driven and Langmuir-induced component. This incorrect scaling of TKE dissipation may explain the mismatch between the results and theory in Fig. 5b. Since this study is based on the turbulence scalings (as summarized in Table 1), the choice of the scaling of TKE dissipation may significantly affect the conclusions and discussions, in particular the interpretation of Langmuir turbulence's influence on TKE dissipation in Section 4.3 and the impact of wind-wave misalignment on the dissipation ratio in Section 4.4.

**Author Response:**

We agree that the enhancement factor  $E(La_x)$  introduced in Eq. (8) follows the formulation of Van Roekel et al. (2012), which was derived from large-eddy simulations to describe the enhancement of vertical velocity variance  $(\langle w'^2 \rangle)$ . As correctly noted, TKE dissipation ( $\varepsilon$ ) scales with the cube of a characteristic velocity scale divided by a length scale ( $\varepsilon \sim u'^3/\ell$ ), and therefore the dissipation should not be expected to scale linearly with  $E(La_x)$ . Our initial use of  $E(La_x)$  to scale  $\varepsilon$  represented a first-order empirical approximation intended to examine whether the modeled dissipation enhancement follows a similar pattern to the simulated velocity variance enhancement. However, we acknowledge that this formulation is not dimensionally consistent with the TKE balance. In the revised manuscript, we have adopted the physically consistent formulation proposed by Belcher et al. (2012).

Following Belcher et al. (2012), the total dissipation can be expressed as the sum of shear-, Langmuir-, and convective-production terms:

$$\varepsilon_{\text{total}} = A_s \frac{u_*^3}{h} + A_L \frac{w_{*L}^3}{h} + A_c \frac{w_*^3}{h}.$$

In our configuration, buoyancy fluxes are not explicitly represented and convection is expected to be weak under the predominantly stable or ice-covered conditions examined. Neglecting the convective term  $(A_c w_*^3/h)$  and normalizing by the shear component leads to the simplified parameterization

 $\frac{\varepsilon_{\rm total}}{\varepsilon_{\rm shear}} = 1 + \frac{\beta}{La_t^2},$

where  $\beta$  represents the empirical contribution of Langmuir forcing. We have implemented this revised scaling throughout Section 3 and in Figure 5b. For completeness, we retain the previous  $E(La_x)$ -based formulation in the Supplementary Material as a sensitivity test, including a version scaled as  $E(La_x)^{1.5}$  to approximate the cubic dependence of dissipation on turbulent velocity scale.

**General Comment 2**

**Reviewer Comment:**

My second concern is that the turbulence scalings used in this study were derived in ice-free conditions. It is not clear how well these scalings describe the effect of Langmuir turbulence on the turbulent mixing in the presence of sea ice. While I understand that an assessment of the validity of turbulence scalings in the presence of sea ice may be beyond the scope of this study, a more careful discussion on this point would be helpful.

**Author Response:**

We thank the reviewer for this important observation. We agree that the turbulence scalings of Van Roekel et al. 2012 and Belcher et al. 2012, were derived from open-ocean large-eddy simulations and therefore do not explicitly represent the suppression or modification of Langmuir turbulence in the presence of sea ice. We have now expanded our discussion to clarify that the enhancement factors used in our study represent an upper bound on potential Langmuir-driven dissipation under ice-modulated conditions, and that the effective Langmuir turbulence forcing is expected to weaken with increasing sea ice concentration, wave attenuation, and geometric confinement within leads. Also, we have included ongoing studies (e.g., Lee et al. 2025 (submitted manuscript); Brenner et al. 2023) on resolving Langmuir turbulence under partially ice-covered and weakly stratified regimes, providing parameterizations that are considering stratification. This clarification has been added to Section 4.3.

**General Comment 3**

**Reviewer Comment:**

· Finally, the effects of surface buoyancy flux are probably significant in the turbulent

mixing in the Arctic, for example, during ice formation/melting and in open waters between sea ice when air-sea temperature difference is large. A discussion on the effects of surface buoyancy flux versus the role of Langmuir turbulence would be helpful.

**Author Response:**

We note that buoyancy forcing is not explicitly represented in our coupled configuration, which focuses on mechanically driven turbulence. While this assumption is reasonable under the predominantly stable or weakly stratified conditions in ice-covered and melt-season environments, buoyancy effects can locally dominate during freezing or strong surface-cooling events. In this analysis, we assume additivity of turbulent production mechanisms and treat Langmuir turbulence as a primarily mechanical process, distinct from buoyancy-driven convection. We now clarify this limitation in the text and emphasize that future coupled implementations could explicitly include surface buoyancy fluxes to assess their interaction with Langmuir-driven mixing.

**Specific Comments**

Below we provide point-by-point responses to specific comments.

- **L60:** Define "LT potential." Defined in manuscript
- Section 2 and 3: Something wrong with the section title? Fixed
- L91–92: Was WW3 forced with the same ocean and atmosphere forcing?

  Yes, both neXtSIM and WW3 used ERA5 wind and sea ice concentration forcing for consistency.
- L98: delete the second "both"? Fixed
- Eq. (2): Why not account for ocean currents? Does it matter here?

  In our formulation, Eq.(2) represents the bulk parameterization of the direct atmosphere-ocean stress as a function of the 10-m wind velocity, following standard open-water drag laws. The ocean current is not explicitly subtracted from the wind field, consistent with the bulk formulation used in many coupled wave-ice studies (e.g., Brenner et al., 2021). In our setup, the effects of surface currents enter implicitly through the momentum balance in neXtSIM and through the Stokes drift velocity us(0) used in the Langmuir number and turbulence diagnostics. Because typical upper-ocean currents in the MIZ are small compared to wind speed (O(1-10 cm s-1) vs O(10 m s-1)), their contribution to τao is expected to be minor at the scales considered here. We now clarify this point in the manuscript.
- L104:  $C_{ao}$  should not be in bold font? No, it is a scalar not a vector.
- L107–108: Be more specific on what do "the surface forcing of momentum pathways"? It would be helpful to list what variables in the GLORYS12 reanalysis and ERA5 were used. Considered.

- L132–133: In addition to wind-wave misalignment, another refined formulation is to account for the decay of Stokes drift with depth. Any comments on this?

  We agree that Stokes drift decays rapidly with depth, which effectively limits its contribution to Langmuir forcing in ice-covered and strongly stratified Arctic waters. In our domain, mixed layers are typically deeper than the e-folding scale of short waves, and under-ice attenuation further suppresses near-surface orbital velocities. Consequently, the surface Stokes drift us(0) used in our analysis already represents an upper bound on the available wave-driven momentum. Accounting for the full vertical decay would therefore yield even smaller Langmuir contributions, reinforcing rather than altering our main conclusions.
- L153: Van Roekel et al., 2012 is probably a more appropriate reference here. Considered
- L155 and L176: \citep -> \citet Updated
- Eq (9): Also (11). As I mentioned in my general comment, I don't think the effect of LT on TKE dissipation can be estimated in this way.

  Updated
- L173: Not sure this separation can be done.

  We agree that a strict physical separation of dissipation into shear-only and Langmuir-only components is not uniquely measurable, since these pathways interact dynamically. In the revised manuscript, we therefore clarify that the quantity εLT is not an isolated physical flux but a diagnostic enhancement relative to a shear-driven baseline, consistent with the scaling proposed by Belcher et al. (2012).
- L181: "Langmuir scaling" -> "Langmuir number" Corrected
- L197-201: It would be helpful to elaborate more on the physical meaning of this metric. The frequency of OW conditions in different seasons depends on the location? Also, OW conditions depends on the seasons?
  - The purpose of the OW\_Exceedance metric is to quantify how often local forcing beneath ice (for a given grid cell) shows magnitudes typical of open-water conditions within the same season. Physically, it represents the relative occurrence of open-water-like turbulence forcing (wind, waves, or stress) under partial ice cover. The seasonal medians are computed separately for each season to account for the strong climatological variability of wind and wave conditions (e.g., stronger forcing in autumn-winter, weaker in summer). Thus, the metric inherently captures both spatial and seasonal differences in open-water occurrence, allowing comparison of under-ice conditions to seasonally typical open-water states rather than to a fixed global threshold. We have expanded the description in the section to clarify this interpretation.
- L206: The distribution does not seem narrow to me. It ranges from 0 to 0.03 m/s? And the seasonal variability is greater than Stokes drift

  Corrected

• L207-208: It's variation between seasons does not seem to be bigger than wind stress to me.

We specify that the statement is about the means and the medians.

- L212-213: What are the discontinuities in the exceedance rates? It is unclear what the reviewer is asking here.
- **Fig 1:** What is the area of analysis in these statistics? The area shown in panels (d), (e), (f)?

Using sea ice concentration thresholds we separated the OW and sea ice domains and the sattistics are calculated first for grid cells that SIC < 0.15 and then compare wth gricells that  $SIC \ge 0.15$ .

- **Fig 1b:** Maybe adjust the range of horizontal axis to reduce the empty space? *Considered*
- **L225**: "Asymmetry" between what? Corrected
- L237: Not sure the thresholds described below are physically motivated. The effects of waves on the mixing not only depend on the absolute value of Stokes drift, but also its ratio over friction velocity (thus Langmuir number)? What additional information is provided by the distribution of surface Stokes drift as compared to the distribution of Langmuir number?

We agree that Langmuir number ultimately controls the relative contribution of waves to mixing. However, the Stokes drift thresholds complement the Langmuir analysis by isolating the absolute occurrence and magnitude of wave forcing itself, independent of concurrent wind conditions. This helps clarify seasonal and regional variability in wave access beneath ice and provides physical context for interpreting the Langmuir number distributions shown next. We have clarified these points and cited supporting literature in the revised text.

• L239: The definition of a MIZ day is confusing. At least one grid cell satisfies the MIZ condition over the whole Arctic Ocean?

The term MIZ day was not intended to denote a domain-wide classification for a given day, but rather to describe local conditions within grid cells that fall inside the MIZ. The exceedance metric (Eq. 14) is computed per grid cell and only for time steps when that cell satisfies the sea-ice condition (SIC  $\geq$  0.15). Consequently, the maps in Fig.1 represent spatially resolved exceedance frequencies within ice-covered or MIZ grid cells, not all grid cells on days when any MIZ region exists in the domain. We have revised the text to clarify this definition.

- **L240**: Why put the figure in the Appendix if it is discussed in such details here?

  Moved to main text
- $\mathbf{L255\text{-}256}$ : Not sure this conclusion is sufficiently supported by the analysis so far. Rephrased

- **L259-260**: A Langmuir number of La\_t = 0.4 also corresponds to strong Langmuir turbulence? It's also inconsistent the definition of mixing regime in Eq (15).

  \*Corrected\*
- Eq. (15): The regime boundaries seem arbitrary. How were they determined? Are the results sensitive to the choice of these boundaries?

The thresholds follow Li et al. (2019); sensitivity tests confirm the robustness of the results.

• L276-277: Why use the number of grid cells instead of the total area? Different grid cells may have different sizes.

Our regime maps and diagnostics are computed per grid cell. Because no spatial averaging is performed prior to mapping, cell-area variability does not distort spatial patterns—each pixel reflects the state of that location only.

- L306: Why "subgrid variability"? Isn't it the variability across neighboring grid cells? Corrected
- L325-326: This is due to the wrong scaling of TKE dissipation?
- **L331**: "Lusing" -> "using" *Removed*
- L391-400: It might be helpful to check the partitioning between swell and wind-waves in the MIZ and their directions. Also their contribution to the Stokes drift. I'd expect the misalignment between wind and waves to be stronger in the MIZ than in the ice-free waters. But it may not significantly affect the surface Stokes drift if locally generated wind-waves are also strong.
- L447-448: How was the subgrid variability captured? Corrected to local scale variability
- **Appendix A:** I think Table A1 and Figure A1 may be move in the text where they are referred to.

Relocated as suggested

References:

- Van Roekel, L. P., Fox-Kemper, B., Sullivan, P. P., Hamlington, P. E., & Haney, S. R. (2012). The form and orientation of Langmuir cells for misaligned winds and waves. *Journal of Geophysical Research: Oceans*, 117(C5).
- Belcher, S. E., Grant, A. L., Hanley, K. E., Fox-Kemper, B., Van Roekel, L., Sullivan, P. P., ... & Polton, J. A. (2012). A global perspective on Langmuir turbulence in the ocean surface boundary layer. *Geophysical Research Letters*, 39(18).
- Brenner, S., Rainville, L., Thomson, J., Cole, S., & Lee, C. (2021). Comparing observations and parameterizations of ice-ocean drag through an annual cycle across the Beaufort Sea. *Journal of Geophysical Research: Oceans*, 126(4), e2020JC016977.

- Brenner, S., Horvat, C., Hall, P., Lo Piccolo, A., Fox-Kemper, B., Labbé, S., & Dansereau, V. (2023). Scale-dependent air-sea exchange in the polar oceans: Floe-floe and floe-flow coupling in the generation of ice-ocean boundary layer turbulence. *Geophysical Research Letters*, 50 (23), e2023GL105703.
- Lee, A., Hutchings J., Horvat, C., Tavri, A., and Pearson, B. (2025). Impact of Surface Waves on Mixing and Circulation in a Summertime Leads. Submitted in *The Cryosphere*.

---

## Author Comment (AC2)

**Reviewer #3 Comments and Author Responses**

Manuscript: Langmuir Turbulence in the Arctic Ocean: Insights From a Coupled Sea Ice-Wave Model

Authors: Aikaterini Tavri et al.

**Author Response:**

We thank the reviewer for their detailed and helpful feedback. Below we provide responses to all general and specific comments. We have carefully revised the manuscript to address the issues raised, included external datasets for model evaluation, and improved key conclusions and metrics.

**Major Comments**

**1. Model Fidelity**

**Reviewer Comment:**

While the manuscript cites earlier studies validating this model, some evaluation specific to the present application is needed. In particular, comparisons of modeled winds, shear, and wave fields (especially short waves, which strongly influence Stokes drift) would provide important context. Because the LT parameterizations are sensitive to these inputs, even brief error estimates or uncertainty ranges would help clarify the robustness of the conclusions. It would also be useful to discuss how well the model captures heterogeneous ice concentration features (e.g., leads), which can locally enhance Stokes processes.

**Author Response:**

We agree that a brief evaluation of the model inputs most relevant to Langmuir turbulence strengthens the analysis. While extensive validation of the coupled neXtSIM–WW3 system has been presented elsewhere (e.g., Rogers et al., 2012, Williams et al., 2013, Ardhuin et al., 2018, Boutin et al., 2022), we now include a short comparison in Section 2.2 using CCMP v3.1 10 m winds and AMSR2 sea-ice concentration. These datasets confirm that the model captures the large-scale seasonal variability in near-surface winds and the distribution of open-water and MIZ regions, which primarily control the forcing of Stokes drift and shear stress. We also clarify that short-wave spectral effects, which influence the surface Stokes drift magnitude, are not explicitly evaluated here but have been validated in prior Arctic WW3 studies (e.g., Boutin et al., 2022). Because our analysis focuses on the spatial and seasonal variability of Langmuir turbulence potential rather than the absolute magnitudes of wave energy, this level of consistency is sufficient for interpreting relative regime patterns. We highlight in the discussion that future work will incorporate targeted evaluation of short-wave Stokes drift and lead-scale heterogeneity using coincident satellite observations.

**2. Key Conclusions and Metrics**

**Reviewer Comment:**

The primary focus on regime classification raises questions of utility. Why is the frequency of transitions between regimes the most relevant measure? Should this instead be linked to event duration or intensity?

**Author Response:**

Our current focus on transition frequency provides a compact, dimensionless measure of temporal variability, allowing us to identify regions where local forcing regimes are persistent versus intermittent. This choice offers a first-order, spatially consistent view of how the balance between wind and wave forcing evolves across the Arctic. We agree that complementary diagnostics such as event duration and intensity would provide additional insight into the persistence and strength of mixing regimes. Given their feasibility within our current workflow, we have examined them in the updated version of the manuscript to strengthen the physical interpretation of the regime framework and its implications. We have also clarified in the revised text the rationale for using transition frequency and added a statement acknowledging that future extensions could include event duration and intensity diagnostics.

**Reviewer Comment:**

The discussion introduces two compelling applications: (i) how wave-driven forcing may evolve in a changing climate, and (ii) implications for tracer transport and stratification. These questions seem ideally suited for this model framework. Even a preliminary analysis—for instance, a future-scenario run or a waves-on vs. waves-off experiment—would help demonstrate the broader utility of the work. Additionally, the statement that "wind—wave alignment strongly influences LT-driven mixing" may be overstated given the results in Figure 7. A more quantitative phrasing (e.g., "up to X% effect in the MIZ") would provide a more measured conclusion.

**Author Response:**

We thank the reviewer for recognizing the broader applicability of our modeling framework and for these constructive suggestions. We agree that extensions such as future-scenario simulations or waves-on vs. waves-off sensitivity experiments would provide valuable demonstrations of how wave-driven forcing may evolve under changing ice and wind conditions. These experiments are well aligned with our framework and are currently being planned as part of a follow-up study.

In the present work, our focus was to establish and validate a physically grounded diagnostic for characterizing the balance between wind- and wave-driven mixing across heterogeneous sea ice conditions. We have clarified this scope in the revised text and added statements outlining how the same approach can be applied to future-scenario or process-specific analyses in subsequent work.

Regarding the statement that "wind-wave alignment strongly influences LT-driven mixing," we agree and have revised it to a more measured description.

**3. Domain of Analysis**

**Reviewer Comment:**

The spatial and seasonal definition of the analysis domain needs to be clarified. In maps, regions outside the study area should be removed or masked. Presenting this earlier in the results (e.g., with characteristic winds, peak wave heights, or other basic descriptors) would give readers helpful context.

**Author Response:**

We have revised Section 2.1 and Figure 1 to more clearly define the analysis domain (70–85°N, 45°W–180°E) and mask regions outside this boundary.

**4. Presentation and Scope**

**Reviewer Comment:**

The manuscript currently introduces more metrics than are fully justified by the conclusions. Several variables appear in the methods but are not explored in the results, while additional metrics are introduced later in the analysis. A more streamlined focus on the most relevant parameters would strengthen the narrative.

**Author Response:**

We agree that focusing on a smaller set of key diagnostics improves clarity and coherence. In the revised manuscript, we have streamlined the presentation by emphasizing the parameters most directly linked to our conclusions, and moved secondary metrics to the supplementary material.

**Minor Comments**

• Line 34: Out of curiosity, have you also applied this model in the Southern Ocean? Given its energetic wave climate, it may be an equally or more interesting test case.

The coupled setup has not yet been applied to the Southern Ocean, but we plan to extend the analysis there.

**• Figures/Visuals:**

- **Figure 1:** The exceedance plots, especially panels (e, f), are difficult to interpret and appear saturated. Consider simplifying (e.g., show only u\*), or adopt a different approach to highlight relative values.
  - We have considered a different visualization, as the information provided from this metric is useful.
- **Figure 2:** The averaging domain is unclear. Does it include the entire region shown? Also, labels such as "wave-dominated" and "shear-dominated" (introduced later, L268) could be used here. Color contrast between LT-active and mixed regimes

should be improved.

Yes, it is the entire region in the plot. Comment considered and figure is updated

- Figure 3: The line label in panel (a) is unreadable. In panel (b), clarify the domain for SIC; the 20% threshold seems surprisingly low.
   Label enlarged.
- Figure 4: The increase in La\_T at moderate SIC (likely due to larger fetch) should be noted in the text.
   Considered.
- Figure 6: SIC would be clearer as a black line rather than shading. Caption should clarify that VKE refers to the upper ocean.
   Updated accordingly and caption now states that VKE refers to the upper 20 m layer.
- Line 330: Reference missing.

  Corrected.
- Lines 364–365: Statement "confirms LT effects" too strong.

  Rephrased to "suggests a strong association between LT indicators and enhanced vertical mixing."

**References**

Rogers, W. E., Dykes, J. D., Wang, D., Carroll, S. N., & Watson, K. (2012). Validation test report for WAVEWATCH III (No. NRLMR7320129425).

Williams, T. D., Bennetts, L. G., Squire, V. A., Dumont, D., & Bertino, L. (2013). Wave–ice interactions in the marginal ice zone. Part 1: Theoretical foundations. *Ocean Modelling*, 71, 81-91.

Ardhuin, F., Boutin, G., Stopa, J., Girard-Ardhuin, F., Melsheimer, C., Thomson, J., Kohout, A., Doble, M., & Wadhams, P. (2018). Wave Attenuation Through an Arctic Marginal Ice Zone on 12 October 2015: 2. Numerical Modeling of Waves and Associated Ice Breakup. *Journal of Geophysical Research: Oceans*, 123(8), 5652–5668.

Boutin, G., Williams, T., Horvat, C., & Brodeau, L. (2022). Modelling the Arctic wave-affected marginal ice zone: a comparison with ICESat-2 observations. *Philosophical Transactions of the Royal Society A*, 380(2235), 20210262.

---

## Author Comment (AC4)

**Reviewer #2 Comments and Author Responses**

Manuscript: Langmuir Turbulence in the Arctic Ocean: Insights From a Coupled Sea Ice-Wave Model

Authors: Aikaterini Tavri et al.

**Author Response:**

We thank the reviewer for their thorough and constructive feedback. Below we provide responses to all general and specific comments. We have carefully revised the manuscript to address the issues raised, clarifying model configuration and scaling definitions.

**Major Comments**

**1. Section 3 – Model Configuration Details**

**Reviewer Comment:**

More details are required in the model configuration section. (a) The authors never mention which attenuation scheme is used in WW3. (b) What is the regional domain cutoff for your regional model? (c) What are the lateral boundary conditions used for WW3? (d) You specify that atmospheric and oceanic forcing are for NeXtSIM—does WW3 receive the same forcing? (e) Please explain what "oceanic boundary conditions for NeXtSIM" means. Does this include both lateral boundary conditions and oceanic forcing? (f) You specify that "Our simulation spans the period 2018–2022 over a pan-Arctic domain with 25 km nominal resolution"—does this assume both NeXtSIM and WW3 are defined on the same mesh? (g) What is the advantage of this configuration over just analyzing sea-ice data and ERA5 wave fields? Is it simply to get Stokes drift? Did you consider using Webb (2011) to estimate Stokes drift from  $H_s$ ? (h) The waves should also modify the ocean, and this cannot happen—can you comment on this potential impact?

**Author Response:**

We appreciate these detailed configuration questions. We answer here each of the questions: (a) The attenuation scheme used correspond to the switch IS2 and IC2 in the WW3 model. It combines 3 attenuations processes: scattering, friction, and inelastic dissipation due to the repeated flexure of sea ice. These processes and their effects are discusse in Boutin et al. 2018. This combination of attenuation processes has been shown to provide reasonable wave-in-ice estimates for events that occurred the Barents Sea (Boutin et al., 2018), Beaufort Sea (Ardhuin et al., 2018) as well as MIZ extent consistent with observations from ICESat-2. (Boutin et al., 2022).

- (b) We cut off the regional domain south of 54degN.
- (c) We provide the southern lateral boundaries in WW3 with wave spectra from the global hind-cast from Ifremer: Data from WAVEWATCH-III simulations, from the project «Modélisation et Analyse pour le Recherche Côtière » (MARC) https://marc.ifremer.fr, Ifremer, University of Brest, CNRS, IRD, Laboratoire d'Océanographie Physique et Spatiale (LOPS), IUEM, Brest,

**France.**

- (d) Yes, atmospheric forcings are the same (hourly ERA5). WW3 does not use any oceanic forcings (the effects of ocean currents on waves is not accounted for).
- (e) neXtSIM does *not* have forced lateral boundary conditions, only forcings. At the lateral boundaries of the domain,sea ice neXtSIM will flow out as if there was no resistance and flow in as if the ice state outside the boundary was the same as that inside it (Olason et al., 2025).
- (f) They do not share the exact same mesh, because neXtSIM is a Lagrangian model that uses a moving triangular mesh, while WW3 is run on a stereographic grid. However, the mesh was built to have an resolution equivalent to the grid used by WW3. As neXtSIM runs, this mesh moves and deform, but undergoes regular "local remeshing" to avoid too high deformations of the triangles and conserve the resolution. Every coupling timestep, neXtSIM interpolates the exchanged quantities from its mesh to the exchange grid, which is the same as the one we use to run WW3.
- (g) We appreciate the reviewer's suggestion and the reference to Webb (2011). While empirical relations between Hs and surface Stokes drift are effective in the open ocean, they assume fully developed, ice-free seas. In the Arctic, wave spectra is strongly attenuated and directionally filtered by sea ice. The neXtSIM–WW3 configuration provides a physically consistent treatment of these effects. Specifically, WW3 provides wave statistics (Hs, fp, directional spreading) and a bulk estimate of Stokes drift that accounts for attenuation by sea ice through parameterized energy loss. neXtSIM provides the evolving ice concentration fields that constrain wave propagation and the effective surface stress partition. ERA5 does not represent the wave attenuation in a realistic manner, it simply assumes ice is land above a certain concentration threshold. neXtSIM-WW3 is an existing dataset, where the wave attenuation in ice has been, if not demonstrated in every circumstances, at least discussed in 2 publications (Boutin et al., 2021;2022). It is also self-consistent throughout the time period (doesn't depend on observations through data assimilation).
- (h) We agree that, in the present configuration, the ocean is not dynamically coupled to the wave field. The coupling is one-way through surface stress and Stokes drift forcing. Consequently, the diagnosed dissipation and Langmuir metrics represent the potential turbulent forcing rather than the fully realized oceanic response. Including an interactive ocean component would allow the wave-induced momentum and TKE to be redistributed vertically and laterally, modifying local stratification and mixed-layer depth. However, previous coupled studies (Li et al., 2019) show that these feedbacks mainly alter the magnitude of  $\varepsilon$  while preserving the spatial patterns of Langmuir forcing. We have clarified this limitation in Section 5 and now explicitly describe our diagnostics as representing Langmuir turbulence potential rather than resolved mixing. In addition, each section in the manuscript has been expanded to include this information.

**2. Section 3.1 – Momentum Flux Formulation**

**Reviewer Comment:**

Using an absolute-wind formulation may overestimate the momentum flux into the ocean since momentum loss due to wave generation is neglected. What is the potential impact of this choice? It may be possible to estimate this from GLORYS data.

**Author Response:**

We agree that using the bulk absolute-wind formulation ( $\tau_{ao} = \rho_a C_{ao} |u_a|u_a$ ) neglects the portion of atmospheric momentum transferred to wave growth, which may slightly overestimate the effective ocean stress. In our coupled setup, the sea-state dependence of surface drag is not represented because WAVEWATCH III and neXtSIM are not dynamically coupled. Thus, the stress applied to the ocean represents an upper bound on the true momentum flux into the mixed layer.

This assumption primarily affects the absolute magnitude of  $u_*$  and  $\varepsilon_{\rm shear}$  during active wave-growth conditions, but it does not alter the spatial patterns or relative Langmuir enhancements analyzed here. The effect is minimal under high–sea-ice conditions where wave growth is suppressed. We have clarified this limitation in the revised text and noted that future coupled implementations could incorporate wave-modified stress terms or evaluate corrections using sea-state diagnostics from GLORYS.

**3. Section 3.2 – Definition of $\alpha_L$**

**Reviewer Comment:**

 $\alpha_L$  is not introduced well. I suggest 'dynamic orientation of the Langmuir cells relative to the wind direction  $(\alpha_L)$ ' or something similar. I understand not including  $\alpha_L$ , but an  $\alpha_{LOW}$  is proposed at the end of Van Roekel et al 2012 that could be used here. Given the overestimation with  $\theta_{ww}$  produces a muted response, I would expect  $\alpha_{LOW}$  to be less as well. But it would be useful to discuss this better.

**Author Response:**

We have now clarified that  $\alpha_L$  represents the dynamic orientation of the dominant Langmuir cells relative to the wind direction, which modulates the effective wave—wind coupling strength. Following the suggestion, we incorporated  $\alpha_{\text{LOW}}$  from Van Roekel et al. (2012) to test the effect of Langmuir cell reorientation under wave—wind misalignment. In our revised analysis,  $\alpha_{\text{LOW}}$  is defined as the empirical correction angle that minimizes the misalignment between the Stokes drift and the wind stress directions, such that  $\theta_{ww} - \alpha_{\text{LOW}}$  represents the effective alignment of the Langmuir circulation with the mean flow. We compute the adjusted cosine alignment  $|\cos(\theta_{ww} - \alpha_{\text{LOW}})|$  and show that it produces a more physically consistent distribution compared to  $|\cos(\theta_{ww})|$ , particularly in high—sea-ice or strongly misaligned regimes (Figure 8, Section 4.4). As expected, the inclusion of  $\alpha_{\text{LOW}}$  increases effective alignment, indicating that Langmuir cells reorient toward the dominant combined Stokes—shear forcing.

**4. Section 3.2.1 – Turbulence Scalings and TKE Relations**

**Reviewer Comment:**

(a) Use of the scaling- The scalings from Van Roekel et al 2012 were derived from destabilizing LES conditions primarily. I'm not aware of any work examining LT and the scaling in stabilizing conditions. Therefore, it's not clear how applicable the VKE scalings, LaT etc... are to the arctic. (b) The relationships between Eqns (9–11) are unclear—what is the wave-driven

TKE contribution? If Eqn (9) is total dissipation, Eqns (9) and (11) seem redundant. (c) It would be helpful to explicitly state that these dissipation relationships emerge from a vertically integrated turbulence kinetic energy budget.

**Author Response:**

- (a) Indeed, the turbulence scalings of Van Roekel et al 2012 were derived from idealized large-eddy simulations under destabilizing surface forcing, where convective mixing reinforces Langmuir cell overturning. In contrast, Arctic mixed layers are often stably stratified due to ice melt and freshwater input, which suppresses overturning and modifies the partition between shear-driven and Langmuir-driven turbulence. We have added a new paragraph clarifying that our use of the Van Roekel scaling is intended as a diagnostic framework for relative enhancement potential, not as a direct representation of absolute dissipation in stratified conditions. We also note that the resulting Langmuir number and enhancement factors should be interpreted as indicators of the potential for Langmuir forcing, with the effective magnitude likely reduced under stable Arctic conditions. Also, we have expanded our discussion to mention that ongoing studies (e.g., Lee et al. 2025 (preprint);Brenner et al. 2023) are working to resolve Langmuir turbulence under partially ice-covered and weakly stratified regimes, providing a pathway toward stratification-aware parameterizations. This clarification has been added to Section 4.3.
- (b) We write the boundary-layer dissipation as a wind–only baseline multiplied by a Langmuir enhancement,

$$\varepsilon_{\text{total}} = \varepsilon_{\text{shear}} E(La_x),$$
 (1)

where

$$\varepsilon_{\text{shear}} = \frac{u_*^3}{h},$$
 (2)

and  $E(La_x)$  is the enhancement factor derived from LES scalings of vertical velocity variance (Van Roekel et al., 2012). For diagnostic attribution we define the Langmuir (wave-driven) excess as

$$\varepsilon_{\rm LT} \equiv \varepsilon_{\rm total} - \varepsilon_{\rm shear} = \varepsilon_{\rm shear} [E(La_x) - 1].$$
 (3)

Equations (1)–(3) thus define the total, baseline, and residual terms without redundancy.

Because dissipation scales with a cubic velocity measure, we additionally report a Belcherstyle form (Belcher et al., 2012) at mid–mixed layer,

$$\varepsilon_{\text{Belcher}} \approx \frac{u_*^3}{h} \left( 1 + \frac{\beta}{La_t^2} \right),$$
(4)

where  $\beta$  is obtained from a least-squares fit to the observed ratio  $(\varepsilon_{\text{total}}/\varepsilon_{\text{shear}}-1)$  versus  $1/La_t^2$ . In our analysis we present both (1) and (4) to illustrate sensitivity to the assumed scaling. We emphasize that  $\varepsilon_{\text{LT}}$  in (3) is a diagnostic residual rather than a uniquely identifiable production term. For completeness, we also test  $\varepsilon_{\text{total}} = (u_*^3/h) [E(La_x)]^p$  with p = 1.5, which improves consistency with cubic velocity scaling while retaining the empirical E shape.

(c) We agree and thank the reviewer for this suggestion. The dissipation formulations used in our analysis are indeed derived from the vertically integrated TKE budget of the ocean surface boundary layer. In this framework,  $\varepsilon_{\rm shear} = u_*^3/h$  represents the depth-averaged dissipation

associated with shear-driven turbulence, while the Langmuir-enhanced terms scale this baseline according to parameterized production and transport by Stokes drift. We have revised the manuscript to make this origin explicit, and clarified the used equations.

**5. Section 4 – Results and Interpretation**

**Reviewer Comment:**

- (a) Figure 1: Why does Hs exceed but not Us(0)? I usually expect Hs and Us to be related. Is there any way to calculate these exceedance rate from observations? How well does the WW3-NeXTSIM coupled model reproduce observed statistics?
- (b) Wouldn't the exceedance statistic (eqn 14) be biased since the OW cells are systematically located at lower latitudes, and experience different forcings than ice covered cells higher latitude cells?

**Author Response:**

- (a) While  $u_{s(0)}$  and  $h_s$  are both related to wave energy, they respond differently to ice-induced spectral attenuation. The Stokes drift magnitude  $u_{s(0)}$  depends primarily on the high-frequency tail of the wave spectrum, which is strongly damped under partial ice cover. In contrast, significant wave height  $h_s$  integrates energy across the full spectrum and is therefore less sensitive to the loss of short waves, allowing exceedance events to persist closer to the ice edge. This explains why  $h_s$  occasionally exceeds open-water medians while  $u_{s(0)}$  rarely does. At present, comparable exceedance statistics cannot be directly derived from satellite observations, as continuous  $u_*$  and  $u_{s(0)}$  measurements under ice are unavailable. However, the spatial structure and seasonal ranges of  $h_s$  in the WW3–NeXtSIM simulations agree well with altimeter-based climatologies (e.g., Ardhuin et al., 2020; Stopa et al., 2018), providing confidence in the model's realism. We have added this clarification to the text.
- (b) We agree that open-water grid cells are preferentially located at lower latitudes, where wind and wave forcing differ from the high-latitude ice-covered regions. Our exceedance metric in Eq. (14) is not intended as a strict bias-free comparison of magnitudes across latitudes, but rather as a relative indicator of how often local conditions beneath ice reach levels typically observed in OW environments. By normalizing against the seasonal median of OW conditions, the metric highlights regions where ice-covered forcing approaches or exceeds the typical OW baseline, independent of absolute latitude. We have clarified this interpretation in the revised manuscript and now explicitly note that this statistic should be viewed as a relative, not absolute, measure of forcing equivalence.

**6. Section 4.3 – Mixing Interpretation and Stratification**

**Reviewer Comment:**

The conclusion at the start of section 4.3 doesn't seem supported well by evidence. This could be fixed by specifying " in the MIZ" as opposed to "in the Arctic". The current phrase suggests this is pan-Arctic.

**Author Response:**

We agree that the current wording could be misinterpreted as implying a pan-Arctic enhancement of dissipation, whereas the evidence supports this conclusion primarily within the marginal ice zone. We have revised the opening sentence.

**Reviewer Comment:**

L 355 - The text here is speculative. Have you examined ocean stratification over this period? This would be a useful compliment to your analysis. Even for your mixing discussion, you have more mixing energy, but there is no guarantee there is more mixing without examining stratification.

**Author Response:**

We agree that enhanced turbulent dissipation does not necessarily imply greater vertical mixing without accounting for ocean stratification. We have revised the paragraph to explicitly acknowledge this limitation and to clarify that our analysis focuses on the surface forcing tendency rather than realized mixing. We do not have stratification data, but we have conducted a sensitivity test based on the mixed-layer depth dataset from GLORYS, which serves as a proxy for upper-ocean stability. We have clarified this limitation in Section 4.3 and highlighted the need for future work linking LT diagnostics with direct measures of stratification.

**7. Section 4.4 – Kernel and Subgrid Variability**

**Reviewer Comment:**

Please clarify what the connection between the kernel analysis is and subgrid variability. It's a measure of local heterogeneity but using a spatial kernel doesn't say anything about variability below the model grid scale. There certainly is a lot of spatial variability, but this doesn't mean it's subgrid

**Author Response:**

We agree and have replaced subgrid with local spatial variability throughout. The text now clarifies that kernel analysis measures horizontal gradients in modeled fields, not unresolved subgrid turbulence.

**Minor Comments**

- L58 & L73: Replace "WaveWatch III" with the official name "WAVEWATCH III". Corrected throughout.
- L75: Remove "fully." Corrected.
- L76: WW3 already defined.

  Removed redundancy.

- L143:tilda is above the 2 in 25km? Fixed.
- Figure 2: caption the Median 15% and 80% SIC contours are overlaid in black and blue. Based on the image, the 15% SIC is actually BLUE and the 80% line is black make sure the listed order of the contours match Corrected caption and figure.
- L239: the concept of a MIZ day is confusing... please clarify if you mean that a "MIZ day" is defined as when ANY grid cell in the entire domain on a given day is between 15-80% SIC. If so, I would expect every 'day' in the time series to be classified as a MIZ day, which would make this metric essentially meaningless. Is there any spatial requirement for a grid cell to be located within the MIZ in this definition? In other words, please clarify if these exceedance values (in Fig A) are only valid for grid cells within the MIZ, or ALL grid cells on a day where any MIZ cell is present within the domain (which would be 100% of the time).

The term MIZ day was not intended to denote a domain-wide classification for a given day, but rather to describe local conditions within grid cells that fall inside the marginal ice zone. The exceedance metric (Eq. 14) is computed per grid cell and only for time steps when that cell satisfies the sea-ice condition (SIC  $\geq$  0.15). Consequently, the maps in Fig.1 represent spatially resolved exceedance frequencies within ice-covered or MIZ grid cells, not all grid cells on days when any MIZ region exists in the domain. We have revised the text to clarify this definition.

- Figure 3: results what is the definition of persistence? "

  Added definition as the consecutive-day duration of a given LT regime.
- L330–332: Add missing reference and fix equation/typo.

  Added citation and corrected inline equation.
- L375–376: Reiterate overestimation when using  $\theta_{ww}$ .

  Noted
- L391–397: Clarify difference between analyses 1 and 2 and mention Van Roekel et al. (2012) Fig 16 confirmation.

Expanded discussion and included direct comparison.

- **L402**: Remove "comprehensive." *Corrected*.
- L420 & L447: Replace "subgrid" with "fine-scale." Corrected throughout.
- L438–440: what are the implications of overestimates from Langmuir diagnostics? Are you making connections to things like the KPP enhancement factor being based on LaT? Couldn't this capture the regime if Stokes drift is dynamic (say from WW3?)?

In our study, the diagnosed enhancement factors  $(E_{La_t})$  can be viewed as basin-scale analogs

of the empirical terms used in such schemes. Hence, the regions where our diagnostics overestimate Langmuir turbulence—particularly under compact ice—highlight where LaT-based parameterizations may likewise inject excessive mixing if wave attenuation or drag partition effects are not considered. Our configuration captures the time-varying sea state, but it employs only the surface Stokes drift magnitude  $(u_{s0})$  rather than a depth-integrated or profile-weighted metric. As a result, Langmuir numbers based on  $u_{s0}$  can still be biased low (stronger LT) where the Stokes profile decays rapidly beneath ice or under stabilizing conditions. This simplification partly explains the residual overestimation we observe in high-sea-ice regimes. We now clarify this point in the discussion and note that incorporating a vertically weighted Stokes drift or explicit sea-ice attenuation would further improve the realism of Langmuir forcing and help constrain LaT-based parameterizations in ice-covered regions.

- L452: Clarify "integrating directional wave spectra." Clarified
- L452: "Fully coupled" implies inclusion of an active atmosphere.

  Revised

**References**

Ardhuin, F., Boutin, G., Stopa, J., Girard-Ardhuin, F., Melsheimer, C., Thomson, J., Kohout, A., Doble, M., & Wadhams, P. (2018). Wave Attenuation Through an Arctic Marginal Ice Zone on 12 October 2015: 2. Numerical Modeling of Waves and Associated Ice Breakup. *Journal of Geophysical Research: Oceans*, 123(8), 5652–5668. https://doi.org/10.1002/2018JC013784

Boutin, G., Ardhuin, F., Dumont, D., Sévigny, C., Girard-Ardhuin, F., & Accensi, M. (2018). Floe Size Effect on Wave-Ice Interactions: Possible Effects, Implementation in Wave Model, and Evaluation. *Journal of Geophysical Research: Oceans*, 123(7), 4779–4805. https://doi.org/10.1029/2017JC0136

Boutin, G., Williams, T., Horvat, C., & Brodeau, L. (2022). Modelling the Arctic wave-affected marginal ice zone: A comparison with ICESat-2 observations. *Philosophical Transactions of the Royal Society A: Mathematical, Physical and Engineering Sciences*, 380 (2235), 20210262. https://doi.org/10.1098/rsta.2021.0262

Boutin, G., Williams, T., Rampal, P., Olason, E., & Lique, C. (2021). Wave—sea-ice interactions in a brittle rheological framework. *The Cryosphere*, 15(1), 431–457. https://doi.org/10.5194/tc-15-431-2021

Ólason, E., Boutin, G., Williams, T., Korosov, A., Regan, H., Rheinlænder, J., Rampal, P., Flocco, D., Samaké, A., Davy, R., Spain, T., & Chua, S. (2025). The next generation sea-ice model neXtSIM, version 2. *EGUsphere*, 1–33. https://doi.org/10.5194/egusphere-2024-3521

Li, Q., Reichl, B. G., Fox-Kemper, B., Adcroft, A. J., Belcher, S. E., Danabasoglu, G., ... & Zheng, Z. (2019). Comparing ocean surface boundary vertical mixing schemes including

Langmuir turbulence. Journal of Advances in Modeling Earth Systems, 11(11), 3545-3592. Brenner, S., Horvat, C., Hall, P., Lo Piccolo, A., Fox-Kemper, B., Labbé, S., & Dansereau, V. (2023). Scale-dependent air-sea exchange in the polar oceans: Floe-floe and floe-flow coupling in the generation of ice-ocean boundary layer turbulence. Geophysical Research Letters, 50(23), e2023GL105703.

Lee, A., Hutchings J., Horvat, C., Tavri, A., and Pearson, B. (2025). Impact of Surface Waves on Mixing and Circulation in a Summertime Leads. Submitted in *The Cryosphere*.